# Short and long sleeping mutants reveal links between sleep and macroautophagy

**Joseph L Bedont[1†], Hirofumi Toda[1†], Mi Shi[1†], Christine H Park[1], Christine Quake[1], Carly Stein[1], Anna Kolesnik[1], Amita Sehgal[1,2]***

[1]Chronobiology and Sleep Institute, Perelman Medical School of University of Pennsylvania, Philadelphia, United States; [2]Howard Hughes Medical Institute, Philadelphia, United States

**Abstract** Sleep is a conserved and essential behavior, but its mechanistic and functional under-pinnings remain poorly defined. Through unbiased genetic screening in *Drosophila*, we discovered a novel short-sleep mutant we named *argus*. Positional cloning and subsequent complementation, CRISPR/Cas9 knock-out, and RNAi studies identified Argus as a transmembrane protein that acts in adult peptidergic neurons to regulate sleep. *argus* mutants accumulate undigested Atg8a(+) auto-phagosomes, and genetic manipulations impeding autophagosome formation suppress *argus* sleep phenotypes, indicating that autophagosome accumulation drives *argus* short-sleep. Conversely, a *blue cheese* neurodegenerative mutant that impairs autophagosome formation was identified inde-pendently as a gain-of-sleep mutant, and targeted RNAi screens identified additional genes involved in autophagosome formation whose knockdown increases sleep. Finally, autophagosomes normally accumulate during the daytime and nighttime sleep deprivation extends this accumulation into the following morning, while daytime gaboxadol feeding promotes sleep and reduces autophagosome accumulation at nightfall. In sum, our results paradoxically demonstrate that wakefulness increases and sleep decreases autophagosome levels under unperturbed conditions, yet strong and sustained upregulation of autophagosomes decreases sleep, whereas strong and sustained downregula-tion of autophagosomes increases sleep. The complex relationship between sleep and autophagy suggested by our findings may have implications for pathological states including chronic sleep disorders and neurodegeneration, as well as for integration of sleep need with other homeostats, such as under conditions of starvation.

*For correspondence:
amita@pennmedicine.upenn.edu

[†]These authors contributed equally to this work

## Introduction

Sleep is a widespread behavior across animals with nervous systems and occupies a significant propor-tion of human life. The importance of sleep is evident in the consequences of its disruption, which range from impaired cognitive performance to serious health problems, and even death in some animal models (*Mignot, 2008*). However, we still have limited understanding of the mechanisms that regulate sleep or the physiological functions served by it.

The fruit fly has been essential to identifying molecular mechanisms regulating sleep. Forward genetic screens in *Drosophila melanogaster* revealed several molecular sleep regulators and effectors later implicated in mammalian sleep. The first was the voltage-gated potassium channel Shaker; its mammalian homolog (Kcna2) was later shown to have corresponding effects in mice (*Cirelli et al., 2005*; *Douglas et al., 2007*). Similarly, we previously identified *redeye,* a nicotinic acetylcholine receptor (nAchR) alpha subunit gene required for sleep maintenance, although cholinergic signaling is typically thought of as wake-promoting, sleep-promoting cholinergic neurons that act through a related nicotinic receptor subunit were subsequently identified in mammals (*Ni et al., 2016*; *Shi et al., 2014*). The power of invertebrate behavioral screening is perhaps best demonstrated by a *tour de force* sleep mutant screen recently conducted in mice; homologs of the two sleep-regulating genes

identified, Nalcn sodium leak channel and Sik3 kinase, were previously linked to sleep in *Drosophila* and *Caenorhabditis elegans,* respectively (*Flourakis et al., 2015*; *Funato et al., 2016*; *van der Linden et al., 2008*). Sleep genes originally identified in flies are increasingly also implicated in human sleep. Both voltage-gated potassium channels and nicotinic acetylcholine receptors were top hits in a genome-wide association study for polymorphisms associated with human sleep duration (*Allebrandt et al., 2013*). And autoantibodies to voltage-gated potassium channels have been found in people with Morvan's syndrome, a neurological disorder associated with insomnia (*Barber et al., 2000*).

The fruit fly has already also proven valuable for interrogating functions of sleep. Proposed functions for sleep across organisms include memory consolidation, both synaptic and metabolic homeostasis, and waste clearance from the brain (*Mignot, 2008*). Sleep promotes memory consolidation in *Drosophila* (*Dag et al., 2019*; *Donlea et al., 2011*), which is also reflected in the deep intertwinement of sleep and memory circuitry (*Haynes et al., 2015*; *Joiner et al., 2006*; *Pitman et al., 2006*; *Sakai et al., 2012*). Meanwhile clearance effects implicated in mammals are inferred from sleep regulation of endocytosis across the *Drosophila* blood-brain barrier (*Artiushin et al., 2018*; *Mestre et al., 2020*). Crucially, whether the varying functions of sleep represent independent outputs of a sleeping brain, or are linked in some manner, is unknown.

A large proportion of waste clearance in cells is mediated by macroautophagy (hereafter, autophagy), which recycles bulk material including protein aggregates and damaged organelles. Different types of autophagy can be induced by stimuli including accumulation of ubiquitinated protein, unfolded protein response, pro-apoptotic signaling, and metabolic stressors including amino acid starvation (*Hale et al., 2013*). Factors involved vary by stimulus, but they converge on a core network of essential proteins that mediate nucleation, expansion, and maturation of an Atg8(+) autophagosome; loading of cargo; and ultimately lysosomal fusion, forming an autolysosome whose acidification and protease activity degrades the cargo (*Hale et al., 2013*). Successful autophagy both remediates toxicity of its cargo and liberates metabolites for reuse by the cell. Autophagy crosstalk with other clearance mechanisms suggest it as a potentially important transducer or effector of sleep. Yet to our knowledge, no direct link between sleep and autophagy has been established.

Here, we report a novel short-sleeping mutant, *argus* (*aus*), derived from a screen of chemically mutagenized flies. The gene responsible for the mutant phenotype encodes an integral membrane protein whose loss in neurons, including in peptidergic subpopulations, reduces sleep by increasing accumulation of undigested autophagosomes. Genetic manipulations that block autophagy upstream of Atg8 recruitment to autophagosomes suppress the *aus* reduced sleep phenotype. Further, in the cases of an independently identified sleep mutant, *blue cheese-58,* and RNAis for several autophagy genes, prominently including *atg1* and *atg8a/b*, blockade of autophagosome production increases baseline sleep. Finally, we show that autophagosomes accumulate during the day, in a manner that can be acutely extended by sleep deprivation or curtailed by enforced sleep. Together, our data suggest that sleep regulates autophagy in a daily sleep:wake cycle, and sustained and/or strong changes in autophagosome level affects sleep amount. This model suggests that autophagy is a promising candidate for coupling sleep to its known functions in the healthy and neurodegenerative brain.

## Results

### *argus* mutants have reduced sleep

As reported previously, we mutagenized newly isogenized *iso31* flies with ethyl methane sulfonate (EMS), generated independent lines, and screened F3 generation flies under 12:12 light:dark cycles for sleep phenotypes (*Shi et al., 2014*). One line that reproducibly showed reduced sleep was named *argus* (*aus*: after the mythological Greek giant who never slept) and subjected to further analysis. *aus* homozygotes showed ~600 fewer minutes of total sleep per 24 hr day compared to *iso31* controls (*Ryder et al., 2004*) controls (p < 0.0001; *Figure 1A–B*). *aus* sleep decrease was primarily driven by inability to sustain sleep, as reflected in a significant decrease in *aus* homozygote sleep bout duration during both day and night (p < 0.0001; *Figure 1C*), while sleep bout number was unchanged during the day and increased at night (p < 0.05; *Figure 1D*). Sleep latency analysis showed that *aus* mutants took a longer time to initiate sleep after lights off at ZT12 than *aus* heterozygotes or controls (p < 0.01; *Figure 1E*). Activity index, locomotor activity per waking minute, was comparable between *aus* mutants and controls (p > 0.05; *Figure 1F*), indicating that *aus* is not a hyperactive mutant. *aus*

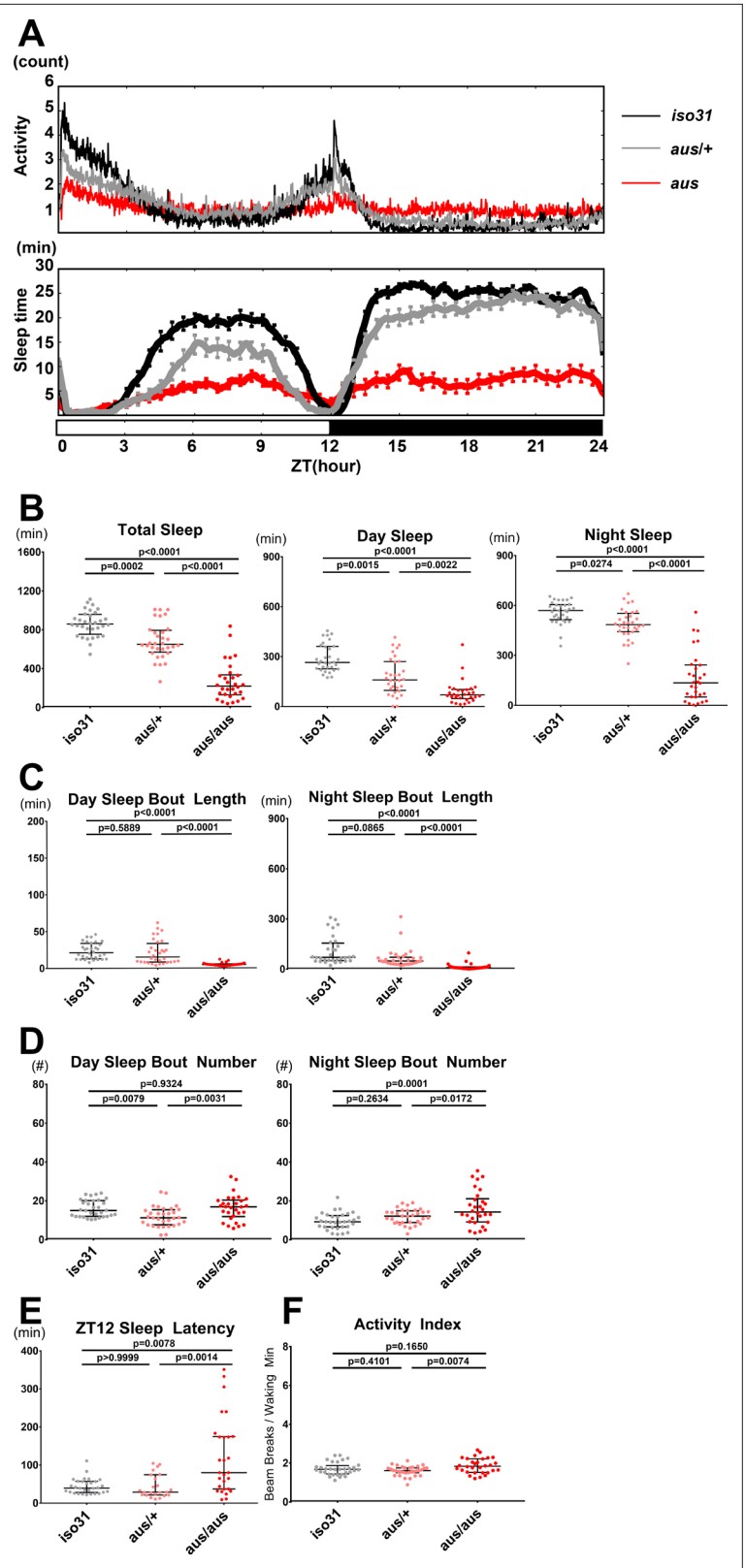

**Figure 1.** Sleep phenotype of *argus* mutants. All sleep metrics were measured under a 12 hr:12 hr light:dark cycle in female iso31 (gray), aus/+ (pink) and aus/aus (red) flies. (**A**) Mean activity (top panel) and sleep (bottom panel) over time during the 24-hr cycle. (**B**) Total sleep amount during the whole 24-hr cycle (left), day (middle), and night (right). (**C**) Mean sleep bout duration during the day (left) and night (right). (**D**) Sleep bout number

*Figure 1 continued on next page*

*Figure 1 continued*

during the day (left) and night (right). (**E**) Latency to first sleep bout after ZT12 lights off. (**F**) Activity index of beam breaks per waking minute over the 24-hr cycle. n = 30–32 (**A–D,F**) or n = 23–30 (**E**); individual flies overlaid with median±interquartiles (**B–F**); Tukey test (B-total+ night,C,F) or Dunn test (B-day,D-E). (**A**).

The online version of this article includes the following source data and figure supplement(s) for figure 1:

**Source data 1.** Sleep Phenotype of Argus Mutants.

**Figure supplement 1.** Circadian rhythms are intact in the *aus* mutant.

**Figure supplement 1—source data 1.** Circadian rhythms are intact in the aus mutant.

---

heterozygotes showed a small decrease in sleep relative to controls (p < 0.001; *Figure 1A–B*), indicating that the *aus* mutation is slightly dominant.

As the circadian clock regulates sleep timing, and some clock mutants show changes in total sleep (*Shaw et al., 2002*), we tested *aus* behavior under constant darkness for a potential circadian phenotype. Most *aus* homozygotes ( > 60%) showed robust locomotor activity rhythms, indicating an intact circadian clock (*Figure 1—figure supplement 1A*). Similarly to other short-sleeping mutants, the overt arrhythmia in ~30% of *aus* homozygotes likely stems from the large reduction in sleep (*Figure 1—figure supplement 1B*; *Shi et al., 2014*). Notably, *aus* homozygotes displayed longer activity episodes than controls in constant darkness (*Figure 1—figure supplement 1A*), consistent with their short sleep under LD conditions.

## Identification of CG16791 as a candidate gene for *argus*

As the original screen selected for recessive mutations on the third chromosome (*Shi et al., 2014*), mapping of the *aus* mutation was initiated by crossing the mutants with a line carrying multiple genetic markers on the third chromosome ($ru^1$ $h^1$ $Diap1^1$ $st^1$ $cu^1$ $sr^1$ $e^s$ $ca^1$). Recombinant progeny were screened for sleep phenotypes and subjected to classical mapping, localizing *aus* distal to *ebony*. We then developed single nucleotide polymorphism (SNP) markers through genomic DNA sequencing of *aus* mutants and the genetic marker line; the *aus* mutation was localized between SNP markers at ~19 and 24 Mb (*Figure 2A*). In parallel, we subjected genomic DNAs from *aus* homozygotes and *iso31* controls to whole-genome sequencing. DNA sequences were aligned with the *Drosophila* Genome Project for SNP calling. While >500,000 polymorphic sites distinguished our stocks from the reference sequence, many SNPs were common to *aus* and the *iso31* control; these were removed from further consideration (*Figure 2B*). In the ~5 Mb region identified by mapping, we found 622 *aus*-specific SNPs, of which 10 led to amino acid changes in nine open-reading frames (ORFs) (*Figure 2B*).

We focused on these ORFs, knocking each down in a pan-neuronal RNAi screen using *elav*-GAL4 driver, and identifying two candidates that produced sleep loss. One was *Neurexin 1* (*nrx1*, cg7050), a synapse assembly molecule that regulates fruit fly sleep (*Tong et al., 2016*). We ruled this candidate out, as the *nrx1* knockout showed no loss of sleep amount compared to control in our hands (*Figure 2—figure supplement 1A*) and it complemented *aus* sleep loss in transheterozygotes (*Figure 2—figure supplement 1B*).

The other candidate gene was *cg16791*, in which *aus* mutant flies have two GC→AT transitions that are predicted to translate to A→V and R→C amino acid substitutions. Supporting its identity as the *aus* locus, pan-neuronal knockdown of cg16791 with elav-Gal4> Dicer and *cg16791* RNAi#1 produced a severe sleep reduction comparable to *aus/aus* mutants (p < 0.0001; *Figure 2—figure supplement 2A,B*). This also suggested the *aus* sleep phenotype is neural in origin. To rule out RNAi off-target effects and confirm this neuron-specificity, we assessed sleep behavior in *cg16791* RNAi #1 and #2 crossed to Nsyb-Gal4> Dicer2 flies. Both pan-neuronal knockdown manipulations resulted in significant decreases in total sleep compared to RNAi and Nsyb-Gal4> Dicer2 controls (p < 0.01; *Figure 2C–D*). These results confirm neuron-dependence of the *cg16791* RNAi sleep phenotype.

Our studies to this point did not address whether *cg16791* acutely regulates adult sleep, or the development of sleep regulatory mechanisms. To address this, we crossed *cg16791* RNAi#1 and #2 to Actin-GeneSwitch (GS)> Dicer2, and conducted sleep experiments in the presence of food supplemented with either the gene switch activating drug mifepristone / RU486 (RU+) or ethanol vehicle control (RU-). Actin-GS> Dicer2+ RNAi#2 flies showed a robust decrease in sleep compared to RNAi and Actin-GS> Dicer2 controls on RU+; however, there was no difference between genotypes on

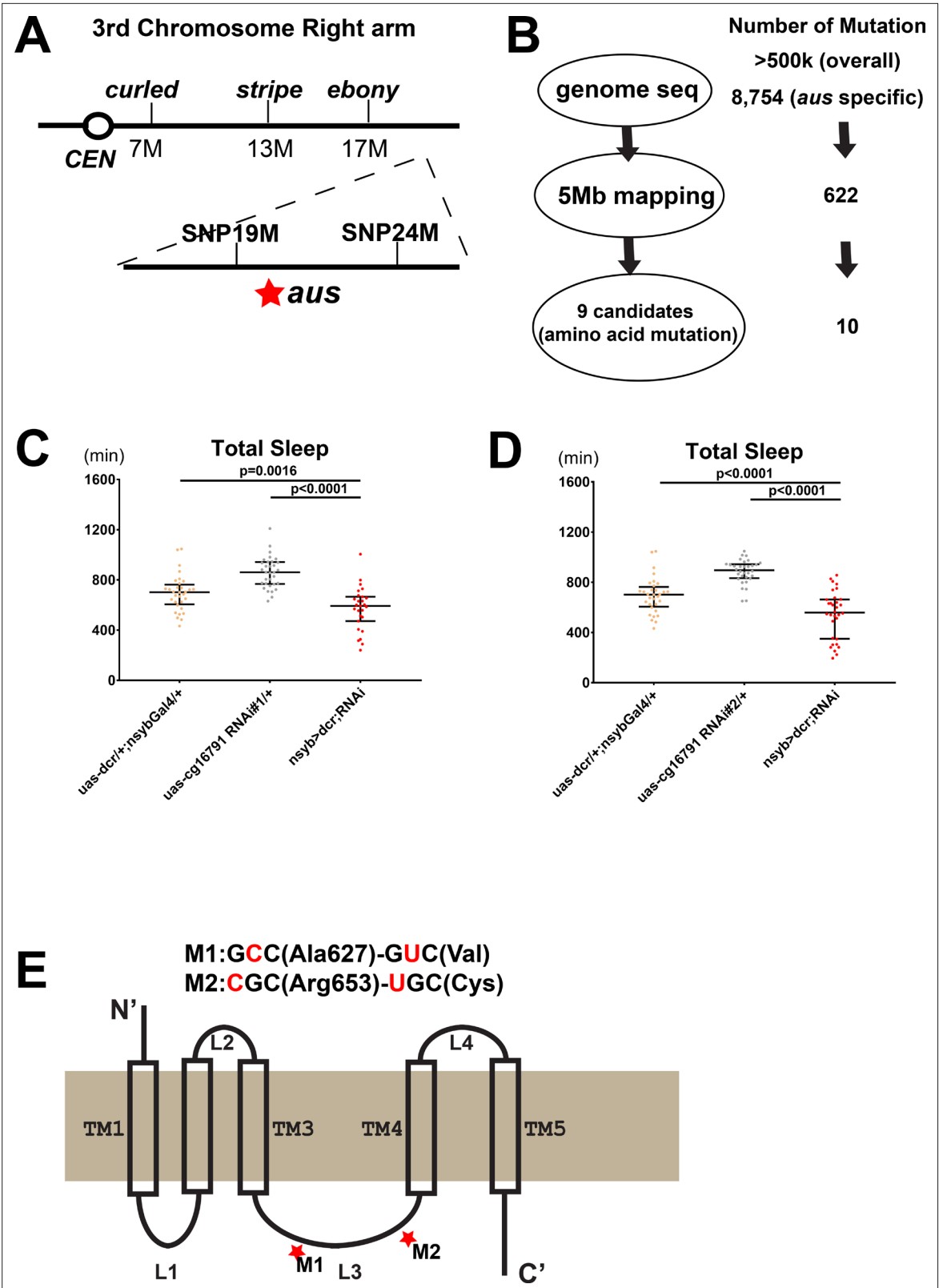

**Figure 2.** Mapping the *argus* sleep phenotype to a single gene: *cg16791*. (**A**) The genomic location of *argus* is indicated as a star within a 5 Mb region on the right arm of the third chromosome, following genetic mapping with visible mutations and SNP markers. (**B**) Schematic of the genome sequencing procedure of *argus* homozygotes with the number of mutations identified in each step listed on the right. The initial alignment revealed more than half a million mutations relative to the published *Drosophila* genome. More than eight thousand mutations remained after removing mutations also found

*Figure 2 continued on next page*

*Figure 2 continued*

in the *iso31* control strain. Factoring in the mapping data (shown in A) and focusing on missense mutations narrowed the number of candidate genes to nine. (**C–D**) Total sleep with *cg16791* RNAi knockdown in females, using pan-neuronal driver nsyb-Gal4, uas-dicer, and either of two independent RNAi lines, compared to RNAi-alone and nsyb-Gal4+ Dcr alone controls. n = 27–32; Fischer's LSD; individual flies overlaid with median±interquartiles. (**E**) Predicted protein of CG16791. Two GC-AT transitions that cause missense mutations in the loop3 region were identified by Sanger-sequencing in *aus* mutants.

The online version of this article includes the following source data and figure supplement(s) for figure 2:

**Source data 1.** Mapping the *argus* sleep phenotype to a single gene: *cg16791*.

**Figure supplement 1.** A mutation in *Nrx1* does not underlie the *aus* reduced sleep phenotype.

**Figure supplement 1—source data 1.** A mutation in Nrx1 does not underlie the aus reduced sleep phenotype.

**Figure supplement 2.** Knockdown of *aus* in adult neurons via RNAi reduces sleep.

**Figure supplement 2—source data 1.** Knockdown of aus in adult neurons via RNAi reduces sleep.

RU- (p < 0.0001; *Figure 2—figure supplement 2C*). Actin-GS> Dicer2+ RNAi#2 flies also showed a within-genotype reduction of sleep on RU+ vs RU- (p < 0.001; *Figure 2—figure supplement 2C*), while the control genotypes did not. This shows that *cg16791* regulates sleep in adulthood. RNAi#1 caused a weak trend toward RU-dependent sleep loss that did not reach significance when crossed with Actin-GS> Dicer2, likely because of weaker knockdown (data not shown); note also that RNAi#2 is predicted to have higher specificity (*Figure 2—figure supplement 2D*).

Having putatively identified the *aus* locus, we took a bioinformatic approach to hypothesize probable structure and function of the largely uncharacterized CG16791 protein isoforms. An unbiased ProDom search of the full-length CG16791 isoform-A reference sequence identified a number of possible transmembrane motifs (*Supplementary file 2*). A more targeted TMPred assessment of known CG16791 isoforms predicted their best-fit membrane topology with a 5-transmembrane structure, and placed the *aus* mutations in an internal loop region between transmembrane helices 3 and 4 (*Figure 2D*; *Supplementary file 2*). This same loop contains a variable region that distinguishes the four known isoforms of CG16791. A Deep-Loc-1.0 analysis predicted that all CG16791 isoforms are targeted to the cell membrane, and perhaps to some extent the ER/Golgi network, driven predominantly by signal sequences in the C-terminus (*Supplementary file 2*; *Almagro Armenteros et al., 2017*). Our bioinformatic analyses are experimentally supported by the isolation of CG16791 isoform-A from membrane fractions of fly head (*Aradska et al., 2015*). Based on these analyses, we speculated that the *aus* substitutions in CG16791's internal loop cause a loss-of-function that underlies its sleep loss phenotype.

## Mutations in CG16791 underlie the *argus* sleep phenotype

To confirm that mutated CG16791 leads to the *aus* sleep phenotype, we performed additional mutant analysis, as well as rescue assays. First, we obtained a P-element insertion allele of *cg16791* that breaks the open reading frame of the gene (hereafter, P1). The P1 allele failed to complement the *aus* mutant. Thus, P1/*aus* trans-heterozygotes had severely reduced total sleep comparable to *aus* homozygotes (p < 0.01; *Figure 3A*), supporting the idea that the *cg16791* mutations in *aus* are causal for the sleep phenotype.

We then used CRISPR/Cas9 to generate a *cg16791* knockout, in which the first exon containing the initiating methionine was replaced with a selectable marker, *Dsred* (*Gratz et al., 2013*; *Figure 3—figure supplement 1A*). As homozygous knockouts were semi-lethal (0.59% survival rate; 2 / 339 flies tested), we could only reliably obtain *cg16791^{KO}* heterozygotes. Southern blot analysis confirmed a single integration of *DsRed* at the *aus* locus (*Figure 3—figure supplement 1B*). Behavioral analysis showed that total sleep in *cg16791^{KO}* heterozygotes is comparable to *aus* heterozygotes, while trans-heterozygotes of *cg16791^{KO}* and *aus* showed a severe reduction in total sleep, similar to *aus* homozygotes (p < 0.0001; *Figure 3B*, *Figure 3—figure supplement 1C*). These *cg16791^{KO}* results further support our mapping of the *aus* EMS allele to loss-of-function of CG16791. However, the *aus* EMS allele maintains function required for survival, as EMS homozygotes are viable while knockout homozygotes are semi-lethal.

The gold standard to confirm that a specific mutation drives a mutant phenotype is through a rescue experiment. We cloned two cDNA forms of *cg16791* under control of a UAS (Upstream Activation

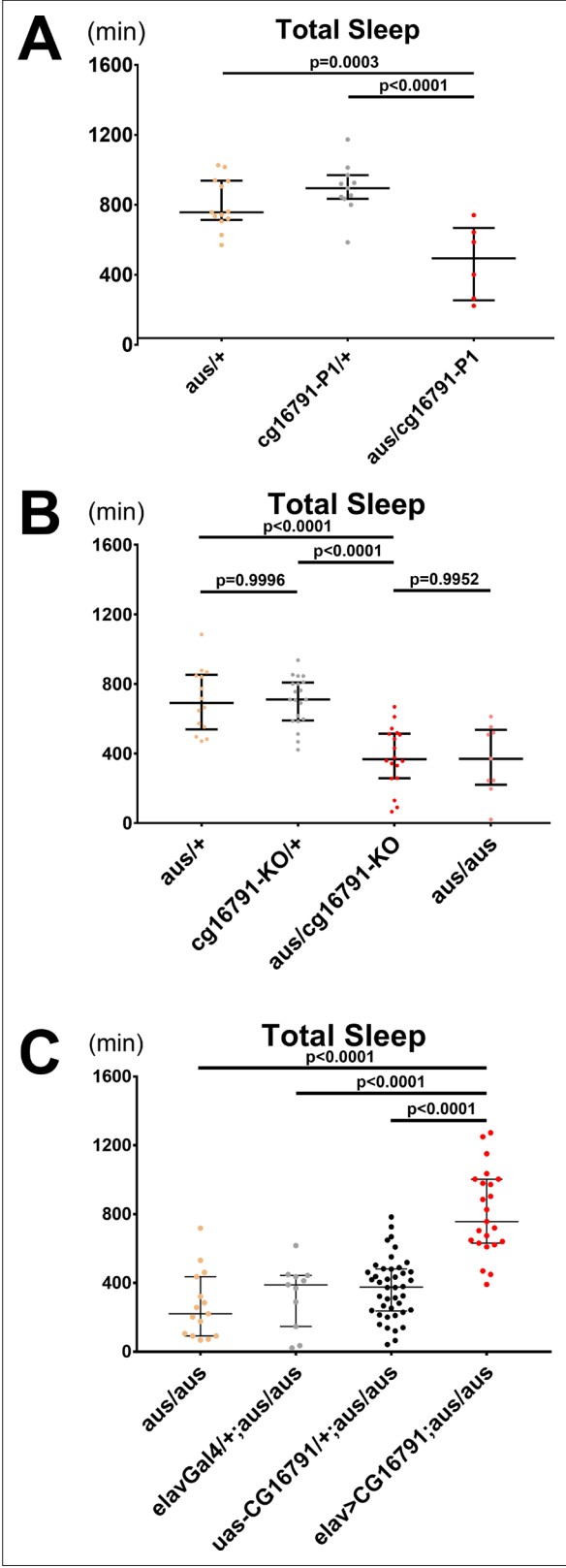

**Figure 3.** CG16791 underlies the *argus* sleep phenotype. (**A**) Transheterozygotes of male *aus* and *cg16791* insertional mutant (**P1**) have reduced total sleep compared to *aus/+* and *cg16791-P1/+* controls. n = 6–13; individual flies overlaid with median±interquartiles; Fischer's LSD. (**B**) Female *cg16791-KO* and *aus* (EMS) transheterozygotes have reduced total sleep compared to *aus* (EMS) or cg16791-KO heterozygotes.

*Figure 3 continued on next page*

*Figure 3 continued*

Transheterozygote total sleep is comparable to *aus* homozygotes. n = 9–20; individual flies overlaid with median±interquartiles; Tukey test. (**C**) Pan-neuronal expression of *uas-cg16791* with elav-Gal4 partially rescues female *aus* homozygote short-sleep, to significantly above *aus*-homozygous Gal4 and UAS controls. n = 11–42; individual flies overlaid with median±interquartiles; Fischer's LSD.

The online version of this article includes the following source data and figure supplement(s) for figure 3:

**Source data 1.** CG16791 underlies the *argus* sleep phenotype.

**Figure supplement 1.** CRISPR-targeting of *argus* to generate a null mutant; supplemental Crispr-KO and full-length rescue data.

**Figure supplement 1—source data 1.** CRISPR-targeting of *argus* to generate a null mutant; supplemental Crispr-KO and full-length rescue data.

Sequence): a UAS-*cg16791* short form beginning with the second methionine, which lacks 51 amino acids at the N-terminus, and a full-length UAS-*cg16791*$^{FL}$ form. Pan-neuronal (*elav*-Gal4) induction of either form effectively rescued sleep in *aus* mutants (p < 0.0001; *Figure 3C*, *Figure 3—figure supplement 1D*), proving that mutations in *cg16791* indeed cause *aus* sleep loss. This result also suggests that the CG16791$^{FL}$ N-terminus is dispensable for its sleep function. For simplicity, only UAS-*cg16791* was used for later experiments.

## Argus functions in dimmed-positive peptidergic neurons to regulate sleep

We next sought to identify the brain region through which *aus* regulates sleep. We first cloned ~2 kb of the *aus* promoter upstream of Gal4 and used the resulting fly line, ausP2k-Gal4, to drive expression of membrane-bound GFP. GFP was expressed broadly in the fly brain, in many cell types including peptidergic neurons of the pars intercerebralis (PI), Kenyon cells of the mushroom body, optic lobe neurons and some lateral neurons (*Figure 4A*). Importantly, P2k-driven *aus* expression rescued *aus/aus* short sleep, indicating that P2k-Gal4 recapitulates the sleep-relevant *aus* expression pattern (p < 0.05; *Figure 4A*).

Based on the prominent representation of neuropeptidergic populations labeled by the Aus2k driver, we suspected that *aus* functions in peptidergic pathways to regulate sleep/arousal behavior. To test this, we assayed for rescue using the peptidergic neuron specific c929-Gal4 driver, which is inserted near the *dimmed* gene, a bHLH transcription factor essential for neuroendocrine cell differentiation, and which appears to express in overlapping cell populations with Aus2k-Gal4 (*Figure 4A–B*, white boxes) (*Hewes, 2003*). c929-Gal4-driven *aus* expression in an *aus* mutant homozygous background partially rescued the short sleep phenotype (p < 0.05; *Figure 4B*), demonstrating that *aus* expression in peptidergic cells contributes to sleep behavior. Furthermore, transheterozygotes for *dimmed* (which have impaired neuropeptidergic neuron function, including in the PI) (*Hewes, 2003*) and *aus* show a synergistic loss of sleep compared to the respective single heterozygotes, suggesting that loss of neuropeptide signaling contributes to the *aus* sleep phenotype (p < 0.01; *Figure 4C*).

## *Aus* mutants show an accumulation of autophagosomes

In investigating the mechanism by which *aus* regulates sleep, we noted that CG16791 was previously identified as a protein that interacts with the cell engulfment receptor Draper (*Fullard and Baker, 2015*). Draper is involved in cell death associated with autophagy, the primary disposal pathway for large-scale cellular waste such as protein aggregates and damaged organelles, and an emergency nutrient source (*McPhee et al., 2010*). Thus, we considered the possibility that AUS plays a role in waste disposal, such as autophagy, within cells. To determine if autophagy is regulated by AUS, we conducted live-imaging experiments in *aus* mutants and *iso31* control flies pan-neuronally by expressing a GFP-mcherry-atg8a fusion protein driven by elav-Gal4 (*Figure 5A*). mCherry red fluorescence, but not GFP green fluorescence, persists under low pH; thus, autophagosomes retain both GFP and mCherry fluorescence, while acidified autolysosomes (autophagosomes that have fused with lysosomes to degrade their cargoes) selectively quench GFP, leaving only mCherry fluorescence (*Mauvezin et al., 2014*).

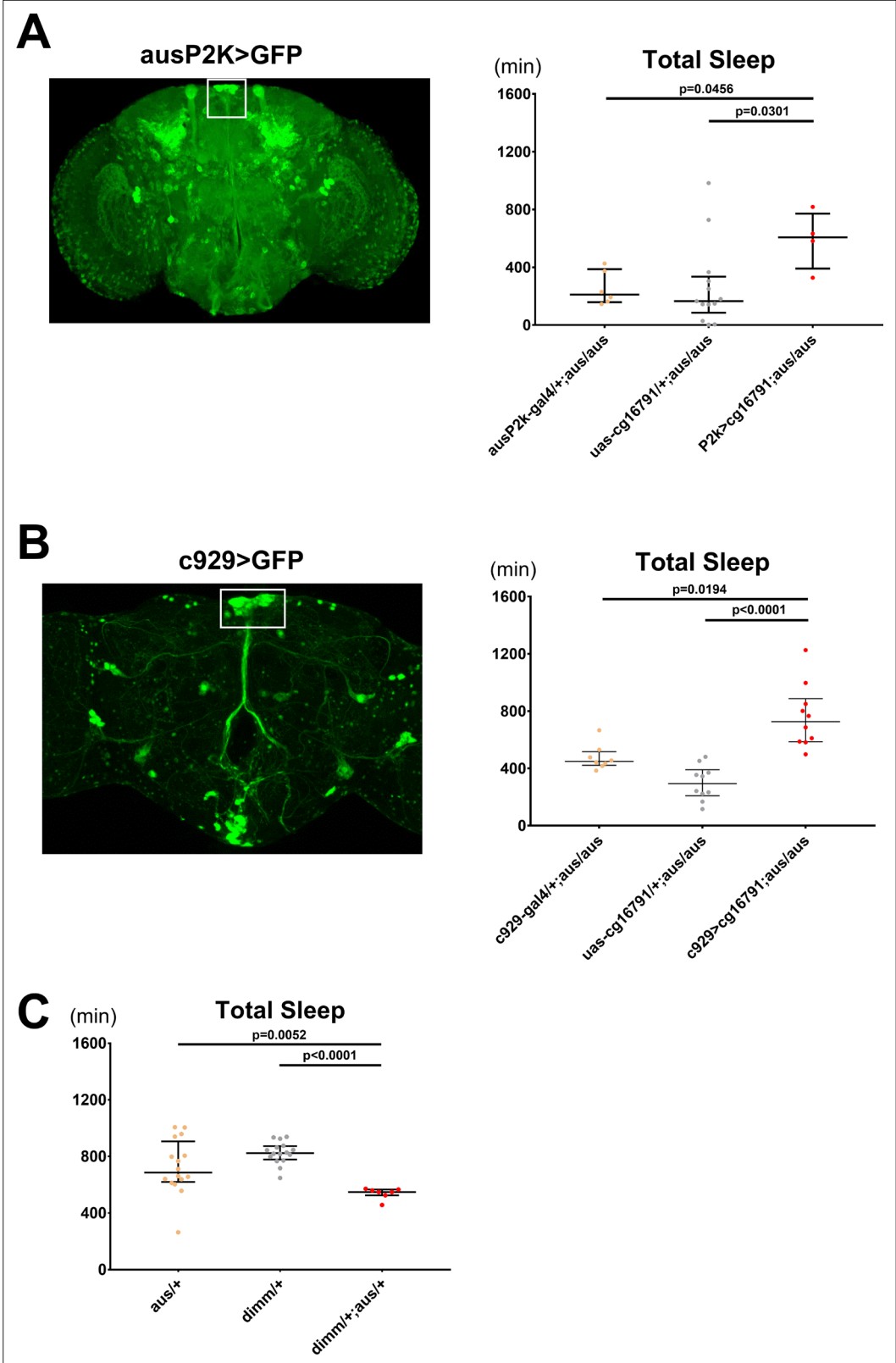

**Figure 4.** Argus functions in dimmed positive neurons to regulate sleep. (**A**) The *aus* promoter region
was subcloned, and a ~ 2000 bp sequence was inserted upstream of Gal4 and used to drive GFP (left).
aus2kGal4 driving *uas-cg16791* partially rescues the short sleep phenotype in female fruit flies (ausP2K, UAS-
*cg16791*, or ausP2K > UAS-cg16791 in *aus/aus* mutant background). n = 4–13; individual flies overlaid with

*Figure 4 continued on next page*

*Figure 4 continued*

median±interquartiles; Fischer's LSD. (**B**) C929-Gal4 (a peptidergic Gal4 line representing Dimmed expression) driving GFP (left). C929 driving *uas-cg16791* expression rescues the short sleep phenotype in male *aus* flies. (c929, UAS-*cg16791*, or c929> UAS-*cg16791* in *aus/aus* mutant background). n = 8–10; individual flies overlaid with median±interquartiles; uncorrected Dunn's test. (**C**) *aus* and *dimm* interact genetically in female transheterozygotes to reduce sleep (*aus/+*, *dimm/+*, and *aus dim* transheterozygotes). n = 7–16; individual flies overlaid with median±interquartiles; uncorrected Dunn's test.

The online version of this article includes the following source data for figure 4:

**Source data 1.** Argus Functions in Dimmed Positive Neurons to Regulate Sleep.

First, we validated a machine learning protocol for identifying neuronal Atg8(+) puncta by comparing autophagy flux in a small cohort of elav-Gal4> UAS-GFP-mCherry-*atg8a* brains dissected from ZT0-2 and incubated in either 2 uM rapamycin or ethanol vehicle in AHL for 2 hr prior to imaging. As expected, given its well-characterized role as a TOR inhibitor and inducer of starvation-dependent autophagy, rapamycin pre-treatment increased total mCherry(+) puncta compared to vehicle control, with no significant difference in either the size of these puncta or the ratio of mCherry+ GFP autophagosomes to mCherry-only autolysosomes (*Figure 8—figure supplement 1A–D*).

We then tested *aus/aus* flies pan-neuronally expressing the same sensor. Neither the number nor the size of all mCherry(+) puncta was significantly different in *aus* mutants compared to controls (p > 0.05; *Figure 5B–C*), but the distribution was significantly skewed toward double-labeled puncta with a reduction in the number of mCherry-alone puncta, suggesting that inefficient lysosomal digestion drives autophagosome accumulation in *aus* mutants (p < 0.001; *Figure 5D*). It is surprising that overall mCherry(+) puncta number is not increased by *aus* blockade of autophagosome degradation; the most likely explanation is a compensatory reduction of autophagy initiation upstream.

## Autophagosome accumulation regulates sleep in *aus* and *blue cheese* mutants

Through an independent project targeted toward identifying links between sleep and neurodegeneration, we discovered a sleep phenotype in the *blue cheese 58* loss-of-function allele (*bchs*). The *bchs* mutant is best known for an autophagy defect that decreases the accumulation of autophagosomes and drives neurodegeneration (*Finley et al., 2003*; *Simonsen et al., 2007*). We found that *bchs* increases sleep compared to *iso31* control (p < 0.01), primarily by lengthening night sleep bouts (p < 0.001) (*Figure 6A*, and *Figure 6—figure supplement 1A–C*). *bchs* also decreases latency to sleep at nightfall (p < 0.0001), suggesting that the sleep gain reflects increased sleep need (*Figure 6—figure supplement 1D*). Activity index was unaffected by either dosage of *bchs*, ruling out defective locomotion as a confound of the sleep gain phenotype (p > 0.05) (*Figure 6—figure supplement 1E*). Many *bchs* sleep phenotypes were recessive in females and dominant in males, suggesting sexual dimorphism.

Overall autophagy is impaired in both *aus* and *bchs* mutants, and yet they have opposing effects on sleep. However, we noticed that while the *aus* mutant increases the accumulation of autophagosomes, likely by blocking their clearance (*Figure 5*), the *bchs* mutant has been shown to decrease the accumulation of autophagosomes, by blocking the maturation of immature Atg5(+) autophagosomes and recruitment of Atg8 (*Sim et al., 2019*). We hypothesized that the opposite changes in the level of Atg8(+) autophagosomes in *aus* and *bchs* mutants drive their respective sleep phenotypes. If true, the *bchs* sleep phenotype should be epistatic to that of *aus*, since the *bchs* blockade of autophagosome maturation is expected to be upstream of *aus* autophagosome accumulation.

To test this, we generated transheterozygous *bchs/+; aus/+* female flies and tested their sleep behavior compared to each single heterozygote. While a single allele of *bchs* had no effect on total sleep amount compared to control *iso31* (*Figure 6A*), it robustly and non-additively suppressed sleep phenotypes of *aus*, rendering the *bchs/+; aus/+* transheterozygotes statistically indistinguishable from *bchs/+* for total, day, and night sleep amount, all of which were higher than *aus/+* (p < 0.05; *Figure 6B*). Activity index was significantly higher in transheterozygotes and *bchs/+* flies compared to *aus/+* flies, suggesting an improvement in *aus* mobility with the addition of *bchs* that does not confound *bchs* suppression of *aus* short-sleep (*Figure 6—figure supplement 1F*). Our findings suggest that the *bchs* sleep phenotype is indeed epistatic to that of *aus*, consistent with their respective effects on the

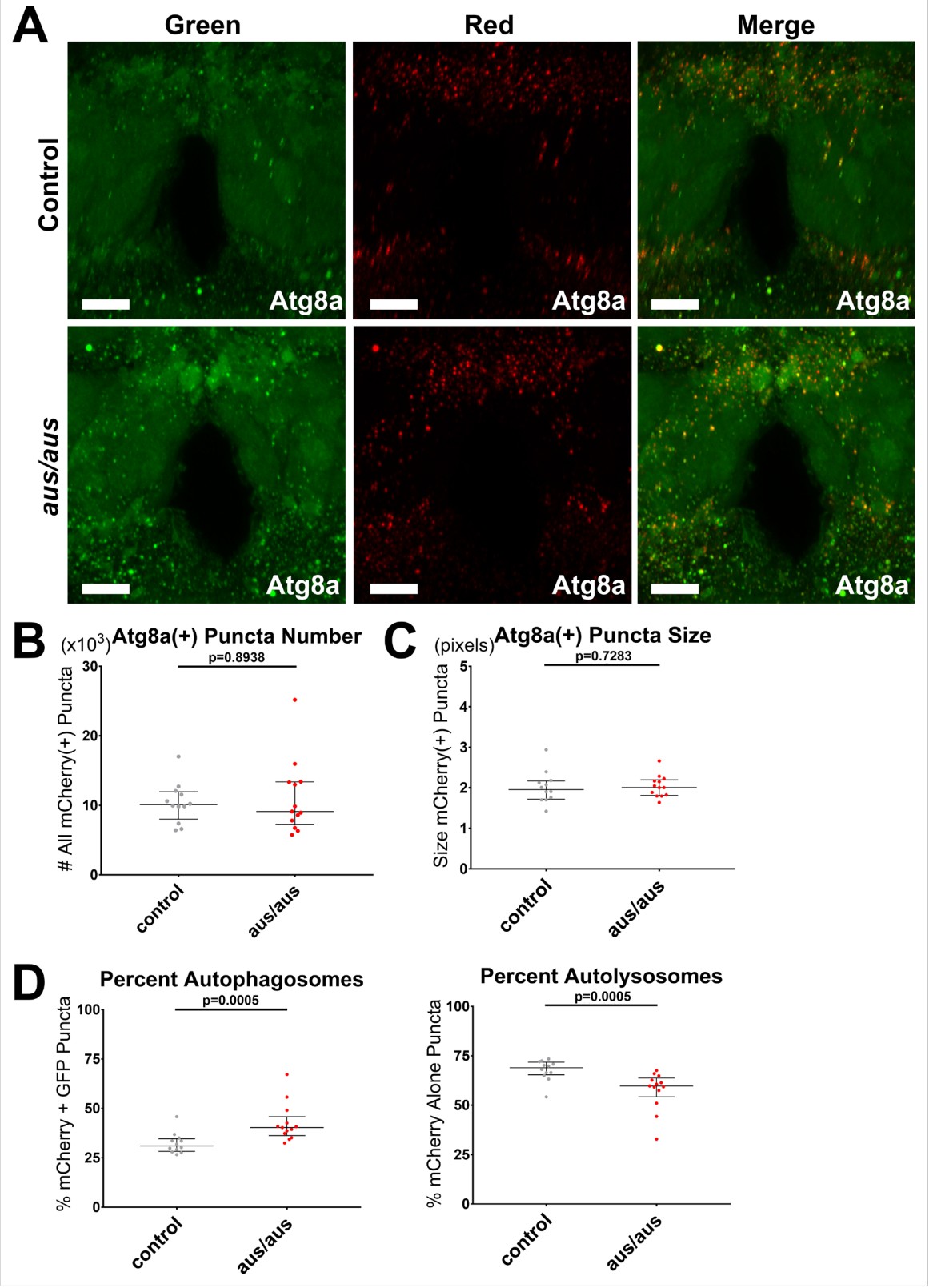

**Figure 5.** The *argus* mutant displays accumulation of autophagosomes. Female *iso31* control and *aus/aus* brains with elav-Gal4> UAS-GFP-mCherry-Atg8a driving pan-neuronal autophagy sensor were live imaged from ZT0-2. mCherry fluoresces in all Atg8a(+) puncta, while GFP fluoresces in autophagosomes and is quenched in autolysosomes. (**A**) Max-projected z-stacks of representative brains showing GFP (left), mCherry (middle), and merged (right) fluorescence. Scale bar = 25 um. (**B**) The number of all neuronal mCherry(+) puncta was similar in both genotypes. (**C**) The size of

*Figure 5 continued on next page*

*Figure 5 continued*

all neuronal mCherry(+) puncta was similar in both genotypes. (**D**) *aus* neuronal mCherry(+) puncta were significantly skewed toward % mCherry+ GFP(+) autophagosomes (left) and away from % mCherry-only(+) autolysosomes (right) compared to control. n = 12–13; individual brains overlaid with median±interquartiles; Mann-Whitney tests.

The online version of this article includes the following source data for figure 5:

**Source data 1.** The *argus* mutant displays accumulation of autophagosomes.

autophagy pathway. This epistatic relationship could not be assessed in trans-homozygous *bchs/bchs; aus/aus* flies because of a lethal interaction.

To confirm that *aus* short-sleep suppression by *bchs* is not due to Bchs roles in other cellular pathways including lysosome trafficking (*Lim and Kraut, 2009*), we assessed whether neuron-specific impairment of autophagosome maturation could rescue short-sleep in *aus* heterozygotes and homozygotes. Thus, we generated homozygous *aus* flies with *elav*-Gal4-driven pan-neuronal RNAi knockdown of *atg5* or *atg7*, both of which are involved in autophagosome maturation and Atg8 recruitment/activation (*Hale et al., 2013*). Both pan-neuronal *atg* RNAis increased sleep on an *aus* mutant background (p < 0.05), driven selectively by increases in night sleep (*Figure 6C–D*). Importantly, neither RNAi increased sleep in flies lacking the *aus* mutation—in fact, *atg7* RNAi decreased sleep in control flies—indicating that the rescue of *aus* did not result from an additive interaction (p < 0.001; *Figure 6— figure supplement 2A,B*). Impaired locomotion cannot explain either sleep rescue phenotype on the *aus/aus* background, as pan-neuronal *atg5* RNAi flies had similar activity index to controls, while the activity index of pan-neuronal *atg7* RNAi flies was intermediate between its controls (p < 0.05; *Figure 6—figure supplement 2C,D*). Much like the *bchs* mutant, *atg5* and *atg7* RNAi also rescue the milder sleep defect of *aus/+* flies (p < 0.001; *Figure 6—figure supplement 2E,F*).

## Blocking autophagosome formation in adulthood increases sleep in wild-type *Drosophila*

While the rescue of *aus* by neuronal knockdown of *atg5* and *atg7* RNAi implicated impaired autophagosome clearance as a mechanism underlying the short-sleep phenotype, we asked why these neuronal knockdowns did not produce a phenotype on their own. The *bchs* sleep gain could be driven by its roles in pathways aside from autophagy, so to rigorously test whether autophagy affects sleep, we conducted a targeted RNAi screen of genes with known links to various steps of autophagy for sleep behavior (*Supplementary file 3*, Tab 1). The use of drug-inducible geneswitch drivers allowed us to restrict manipulations to adulthood.

We first screened with pan-neuronal nsyb-geneswitch+ uas-dicer on RU+ food, and identified five RNAis for four genes that significantly increased sleep (p < 0.05; *Figure 7A*). These included upstream regulators that couple autophagy to starvation (Atg1, 2 RNAi's), unfolded protein response (Bip), and ecdysone signaling (Daor) (*Figure 7—figure supplement 1A,B,D,E*; *Hale et al., 2013*). The remaining gain-of-sleep hit was Atg10, an E2 ligase-like enzyme involved in autophagosome vesicle expansion (*Figure 7—figure supplement 1C*; *Hale et al., 2013*).

To accomplish broad and strong knockdown of autophagy genes, we repeated the same screen with actin-geneswitch. As expected, this approach yielded more gain-of-sleep hits (*P* < 0.05; *Figure 7B*), including several autophagosome maturation proteins: two distinct RNAis encoding Atg8b (one of two ubiquitin-like homologs that label mature autophagosomes), and single RNAis for Atg12 and Atg7 (*Figure 7—figure supplement 1H,I,J,K*; *Hale et al., 2013*). Notably, the *atg7* hit was the same allele used to rescue *aus*, suggesting that sleep loss from its knockdown with elav-Gal4 in the absence of *aus* reflects dosage and/or developmental compensation effects (*Figure 6* and *Figure 6—figure supplement 1B*). Additional RNAi hits encoded proteins involved in autophagy initiation by multiple pathways (Aduk, Atf6, Daor, Dram, Wacky); autophagosome nucleation (Atg14, Dor); and facilitating autophagosome-autolysosome fusion (also Dor) (*Figure 7—figure supplement 1F,G,L,M,N,O,P*; *Hale et al., 2013*; *Lindmo et al., 2006*; *Montagne, 2016*). All of these hits consistently increased sleep, in the case of Dor likely because of an epistatic effect on autophagosome nucleation (*Lindmo et al., 2006*).

To validate these results, we back-crossed our highest confidence hits (*atg1* RNAi's #1,4 and *atg8b* RNAi's #1,2) to *iso31* and closely assessed sleep with crosses to nsybGS and actinGS (respectively).

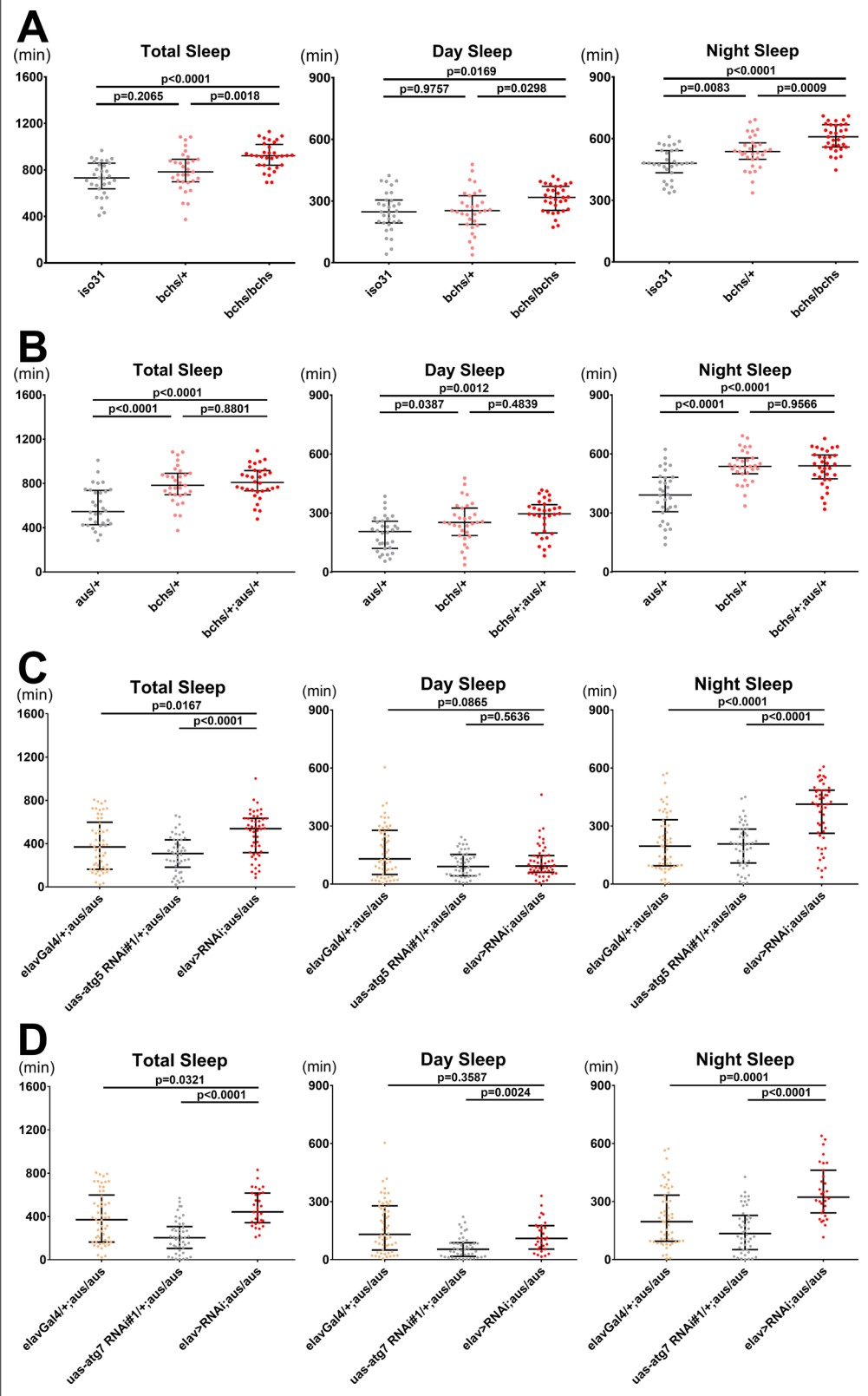

**Figure 6.** Blocking autophagosome production rescues the short sleep phenotype of the *argus* mutant. (**A**) Total, day, and night sleep were measured under 12 hr:12 hr light:dark in *iso31* control, *bchs/+*, and *bchs/bchs* female flies. n = 31–32; individual flies overlaid with median±interquartiles; Tukey tests. (**B**) Total, day, and night sleep were measured under 12 hr:12 hr light:dark in *aus/+*, *bchs/+*, and *bchs/+; aus/+* transheterozygous female flies.

*Figure 6 continued on next page*

*Figure 6 continued*

n = 31–32; individual flies overlaid with median±interquartiles; Tukey tests. (**C–D**) Total, day, and night sleep were measured under 12 hr:12 hr light:dark in elav-Gal4/+, UAS-RNAi/+ and elav-Gal4> UAS RNAi female flies in *aus/ aus* mutant background. RNAi's used were atg5 RNAi#1 (**C**) and atg7 RNAi#1 (**D**). n = 46–54 (**C**) or n = 31–54 (**D**); individual flies overlaid with median±interquartiles; uncorrected Dunn's tests.

The online version of this article includes the following source data and figure supplement(s) for figure 6:

**Source data 1.** Blocking autophagosome production rescues the short sleep phenotype of the *argus* mutant.

**Figure supplement 1.** Effects of *aus* and *bchs* on sleep consolidation, latency, and activity index.

**Figure supplement 1—source data 1.** Effects of aus and bchs on Sleep Consolidation, Latency, and Activity Index.

**Figure supplement 2.** Rescue of *aus/atg7* RNAi.

**Figure supplement 2—source data 1.** Rescue of *aus* mutants by *atg5/atg7* RNAi.

Both *atg1* RNAi crosses increased total sleep (p < 0.0001; *Figure 7C–D*). The nsybGS> dcr,atg1 RNAi#1 increased sleep largely RU-dependently, while nsybGS> dcr,atg1 RNAi#4 increased sleep largely RU-independently, suggesting a leaky GS/RNAi combination (*Figure 7C–D*). Neither total mean bout length nor total bout number was significantly increased in either cross (*Figure 7—figure supplement 2A,B,G,H*). But both *atg1* RNAi crosses had many flies with massive single night-time sleep bouts, and on RU+ food we found consistently longer night (p < 0.01) and longest (p < 0.0001) sleep bout lengths, with significantly lower night bout number (p < 0.05), suggesting that consolidation of night sleep drives overall sleep gain in these flies (*Figure 7—figure supplement 2A,B,C,G,H,I*). Sleep latency at nightfall was consistently decreased in both *atg1* RNAi crosses on both foods (p < 0.01), with an even stronger decrease on RU+ vs RU- (p < 0.001; *Figure 7—figure supplement 2D,J*). Food-independent increased waking activity in *atg1* knockdowns excludes the possibility that sleep increases are derived from sickness (p < 0.05; *Figure 7—figure supplement 2E,K*). Finally, qPCR quantification confirmed knockdown of *atg1* by our RNAi alleles in actinGS> dcr,atg1 RNAi flies (p < 0.05; *Figure 7—figure supplement 2F,L*).

Both actinGS> dcr,atg8b RNAis robustly and RU-dependently increased night sleep, but only RNAi#2 increased total sleep and day sleep after back-crossing (*Figure 7E–F*). Both *atg8b* RNAis RU-dependently increased mean sleep bout length (p < 0.05), driven disproportionately by longest bout (p < 0.05; *Figure 7—figure supplement 2M-O,S-U*). Sleep latency at nightfall was marginally decreased on RU+ food compared to RU- in actinGS> dcr,atg8b RNAi flies, but not control genotypes (p < 0.05; *Figure 7—figure supplement 2P,V*). Neither *atg8b* RNAi cross had significantly different waking activity compared to both controls on either RU+ or RU- food (*Figure 7—figure supplement 2Q,W*). As qPCR did not consistently detect atg8b even in control fly extracts, suggesting very low expression, *atg8b* RNAis may effect their sleep gain by knockdown of *atg8a* through conserved sequences. This was supported by qPCR quantification of *atg8a* cDNA in actinGS> dcr,atg8b RNAi flies (p < 0.05; *Figure 7—figure supplement 2R,X*).

In sum, pan-neuronal *atg1* and whole-fly *atg8* knockdown phenotypes largely recapitulate the key hallmarks of *bchs* phenotypes: (1) sleep gain disproportionately driven by night sleep, (2) sleep consolidation, and (3) decreased sleep latency at nightfall (*Figure 6* and *Figure 6—figure supplement 1*). This supports our attribution of the *bchs* sleep phenotype to its autophagy effects and, more generally, the sleep promoting effects of blocking autophagosome formation. RU-dependence of many phenotypes demonstrates that perturbing autophagosome formation in adulthood is sufficient to drive changes in sleep.

## Sleep negatively regulates autophagosome formation in *Drosophila*

Our findings above indicated that autophagy, in particular autophagosome levels, regulate sleep amount. To determine whether sleep, in turn, regulates autophagosome accumulation, we first live-imaged neuronal autophagy flux in brains of flies carrying elav-Gal4> UAS-GFP-mCherry-*atg8a* at ZT0-2 and ZT12-14 (*Figure 8A*). In the early night, there were more total mCherry(+) puncta than in the early day, with no significant difference in the size of mCherry(+) puncta or the ratio of mCherry+ GFP autophagosomes to mCherry-only autolysosomes, suggesting a potential role for sleep:wake state in regulating the production of autophagosomes (p < 0.05; *Figure 8B–D*). However, this experiment left

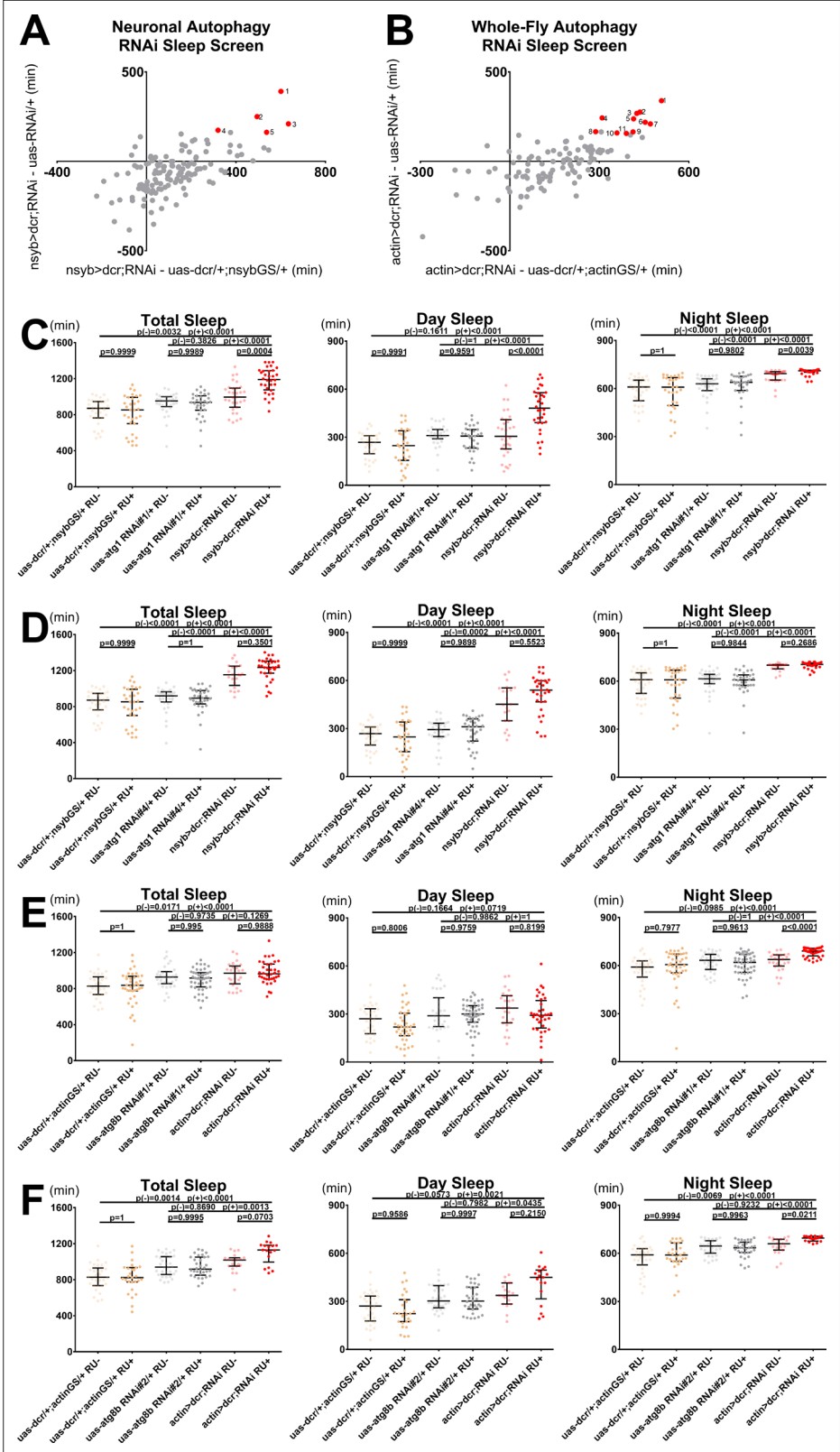

**Figure 7.** Blocking neuronal or whole-fly autophagosome formation increases sleep. (**A**) Difference in first-pass population median sleep on RU+ food for a range of female nsybGS> dcr;autophagy-RNAi crosses compared with nsybGS> dcr control (x-axis) and RNAi control (y-axis). Red, numbered dots indicate significant hits that passed all validation steps: (1) *bip* RNAi#3; (2) *atg1* RNAi#4; (3) *daor* RNAi#1; (4) *atg1* RNAi#1; (5) *atg10* RNAi#3. N = 133

*Figure 7 continued on next page*

*Figure 7 continued*

viable crosses shown; n = 3–16 flies per group for each first-pass experiment. (**B**) Difference in first-pass population median sleep on RU+ food for a range of female actinGS> dcr;autophagy-RNAi crosses compared with actinGS> dcr control (x-axis) and RNAi control (y-axis). Red, numbered dots indicate significant hits that passed all validation steps: (1) *dor* RNAi#2; (2) *atf6* RNAi#1; (3) *atg8b* RNAi#2; (4) *wacky* RNAi#2; (5) *atg8b* RNAi#1; (6) *atg7* RNAi#1; (7) *daor* RNAi#1; (8) *atg14* RNAi#3; (9) *dram* RNAi#2; (10) *aduk* RNAi#3; (11) *atg12* RNAi#2. N = 106 viable crosses shown; n = 3–16 flies per group for each first-pass experiment. See *Supplementary file 3* for details on first-pass screen and *Figure 7—figure supplement 1* for combined first/second pass sleep data for significant hits, for the screens shown in both 7A and 7B. (**C–F**) Total (left), day (middle), and night (right) sleep in GS> dcr;RNAi, GS> dcr control, and RNAi control female flies on both RU+ and RU- food. All data shown as individual flies overlaid with median±interquartiles; p(-) indicates RU- p-values and p(+) indicates RU+ p-values. (**C**) nsybGS> dcr;atg1-RNAi#1: n = 31–32; Steel-Dwass test (total,night) and Tukey test. (day). (**D**) nsybGS> dcr;atg1-RNAi#4: n = 21–32; Steel-Dwass tests. (**E**) actinGS> dcr;atg8b-RNAi#1: n = 25–47; Steel-Dwass test (total,night) and Tukey test. (day). (**F**) actinGS> dcr;atg8b-RNAi#2: n = 17–32; Steel-Dwass tests.

The online version of this article includes the following source data and figure supplement(s) for figure 7:

**Source data 1.** Blocking Neuronal or Whole-Fly Autophagosome Formation Increases Sleep.

**Figure supplement 1.** Validated hits from autophagy RNAi screens.

**Figure supplement 1—source data 1.** Validated hits from autophagy RNAi screens.

**Figure supplement 2.** *atg1* and *atg8b* RNAi additional sleep metrics, activity index, and validation of knockdown.

**Figure supplement 2—source data 1.** *atg1* and *atg8b* RNAi additional sleep metrics, activity index, and validation of knockdown.

---

ambiguous whether sleep contributed to the observed day/night effect. To address this, we mechanically sleep-deprived (SD) flies of the same genotype overnight for at least 12 hr, and compared autophagy flux in SD vs control flies at ZT0-2 (*Figure 8E*). SD flies had significantly more total mCherry(+) puncta compared to controls, with no significant difference in the size of mCherry(+) puncta or the ratio of mCherry+ GFP autophagosomes to mCherry-only autolysosomes (*Figure 8F–H*).

To complement our SD data and mitigate possible confounding effects from the stress of mechanical perturbation, we also assayed effects of increased sleep on autophagy. We flipped flies onto food laced with either gaboxadol or water vehicle at ZT0-1, and compared ZT1-12 sleep and ZT12-14 autophagy flux in the same flies (*Figure 8I*). As previously reported, gaboxadol treatment markedly increased sleep (*Figure 8—figure supplement 1E*; *Berry et al., 2015*). Gaboxadol flies had significantly fewer total mCherry(+) puncta compared to controls, with no significant difference in either the size of mCherry(+) puncta or the ratio of mCherry+ GFP autophagosomes to mCherry-only autolysosomes (*Figure 8J–L*).

These data showing that wake increases and sleep decreases autophagosome number in wild-type fly neurons (*Figure 8*) were unexpected because they could be interpreted as sleep-promotion by autophagosomes, while our mutant and RNAi data indicate that high neuronal autophagosome number decreases sleep and low neuronal autophagosome number promotes sleep (*Figures 1 and 5–7*). As discussed below, we believe that the phenotypes of the mutants/RNAis reflect sustained high or low levels of autophagosomes not seen during a normal daily cycle (*Figure 9*).

## Discussion

Using a forward genetic screen, we identified a novel neural regulator of both sleep and autophagosomal clearance: *argus* (*cg16791*) (*Figures 1–3* and *5*). Autophagosomes accumulate in *aus* mutants, likely due to impaired lysosomal clearance, and multiple genetic manipulations that disrupt whole-fly or neuronal autophagosome formation rescue *aus* mutant sleep (*Figures 5 and 6*). A link between sleep and autophagy is further supported by our finding of additional sleep phenotypes upon downregulation of components of autophagic pathways (*Figures 6 and 7*).

*Drosophila* is a powerful model for the use of unbiased approaches to identify the molecular basis of a physiological process of interest. Indeed, the molecular basis of the circadian clock was determined largely through forward genetic screens of the type used to isolate *aus* (*Dubowy and Sehgal, 2017*). We and others have employed a similar forward genetics toolkit to discover sleep-regulating genes (*Dubowy and Sehgal, 2017*). These genes have provided insight into mechanisms that control

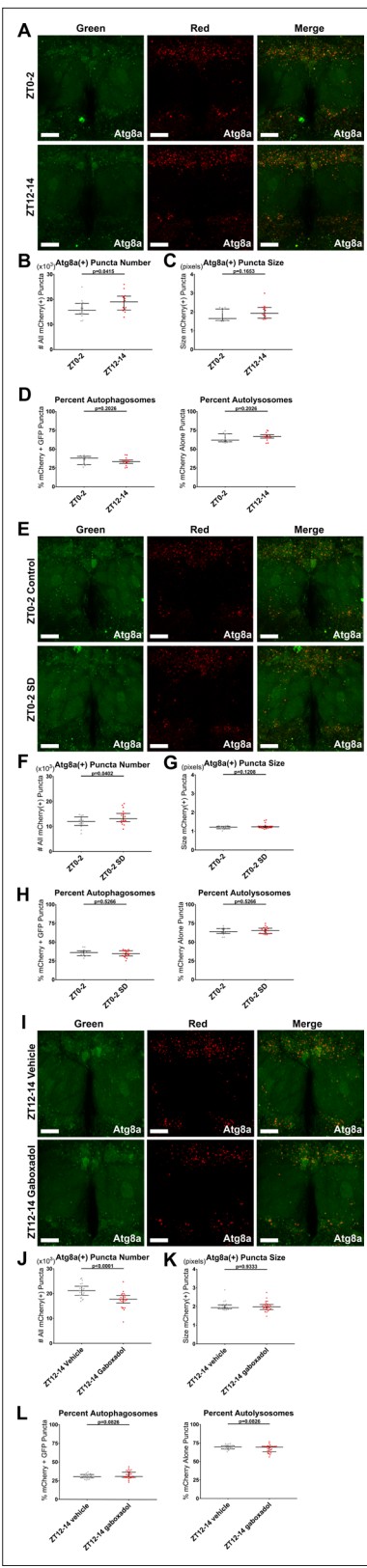

**Figure 8.** Sleep regulates autophagosome production. elav-Gal4> UAS-GFP-mCherry-Atg8a flies expressing pan-neuronal autophagy sensor were live imaged as follows. All quantification shows individual brain values

*Figure 8 continued on next page*

*Figure 8 continued*

overlaid with population median±interquartiles. (**A–D**) ZT0-2 or ZT12-14. n = 15; Student's t-tests. (**E–H**) ZT0-2 after either a control night of unchallenged sleep or at least 12 hr of mechanical sleep deprivation (SD) beginning at the prior ZT12. n = 13–20; Student's t-tests. (**I–L**) ZT12-14 after either a control day of feeding with vehicle or at least 11 hr of feeding with 0.1 mg/mL gaboxadol that verifiably and markedly increased daytime sleep, beginning at the prior ZT0-1. n = 25–26. (**A,E,I**) Max-projected z-stacks of representative brains showing GFP (left), mCherry (middle), and merged (right) fluorescence for ZT time comparison (**A**), control vs SD (**E**), or vehicle vs gaboxadol (**I**). Scale bars = 25 μm. (**B,F,J**) The number of all neuronal mCherry(+) puncta was higher at nightfall than daybreak (**B**), elevated at daybreak by 12 hr overnight SD (**F**), and depressed at nightfall by 12 hr daytime of gaboxadol-induced sleep (**J**). (**C,G,K**) The size of all neuronal mCherry(+) puncta was unaffected by ZT time, SD, and gaboxadol. (**D,H,L**) The percentage of neuronal mCherry(+) puncta that are mCherry+ GFP(+) autophagosomes (left) and mCherry-only(+) autolysosomes (right) was unaffected by ZT time, SD, and gaboxadol.

The online version of this article includes the following source data and figure supplement(s) for figure 8:

**Source data 1.** Sleep regulates autophagosome production.

**Figure supplement 1.** Validation of the Ilastik algorithm for scoring autophagy and the gaboxadol effect on sleep.

**Figure supplement 1—source data 1.** Validation of the Ilastik algorithm for scoring autophagy and the gaboxadol effect on sleep.

daily sleep amount, but to date, they have not suggested functions of sleep. For the most part, the genes identified encode neuromodulators or regulators of neural excitability, which are likely required to modulate brain activity in response to homeostatic sleep need (*Cirelli et al., 2005*; *Koh et al., 2008*; *Shi et al., 2014*). The generation of sleep need is presumably linked to sleep function, but the nature of this remains elusive. The *aus* sleep mutant is unique, in that the mechanisms underlying its loss of sleep are likely relevant for sleep function (discussed below).

Our finding that *aus* regulates autophagy is consistent with its expression pattern spatially, temporally, and even intracellularly. *aus* is temporally elevated during the embryo cellularization and late larval / early pupal stages of fruit fly development, times of enhanced developmental autophagy. Indeed, high autophagy during the latter stage provided the first observation of the

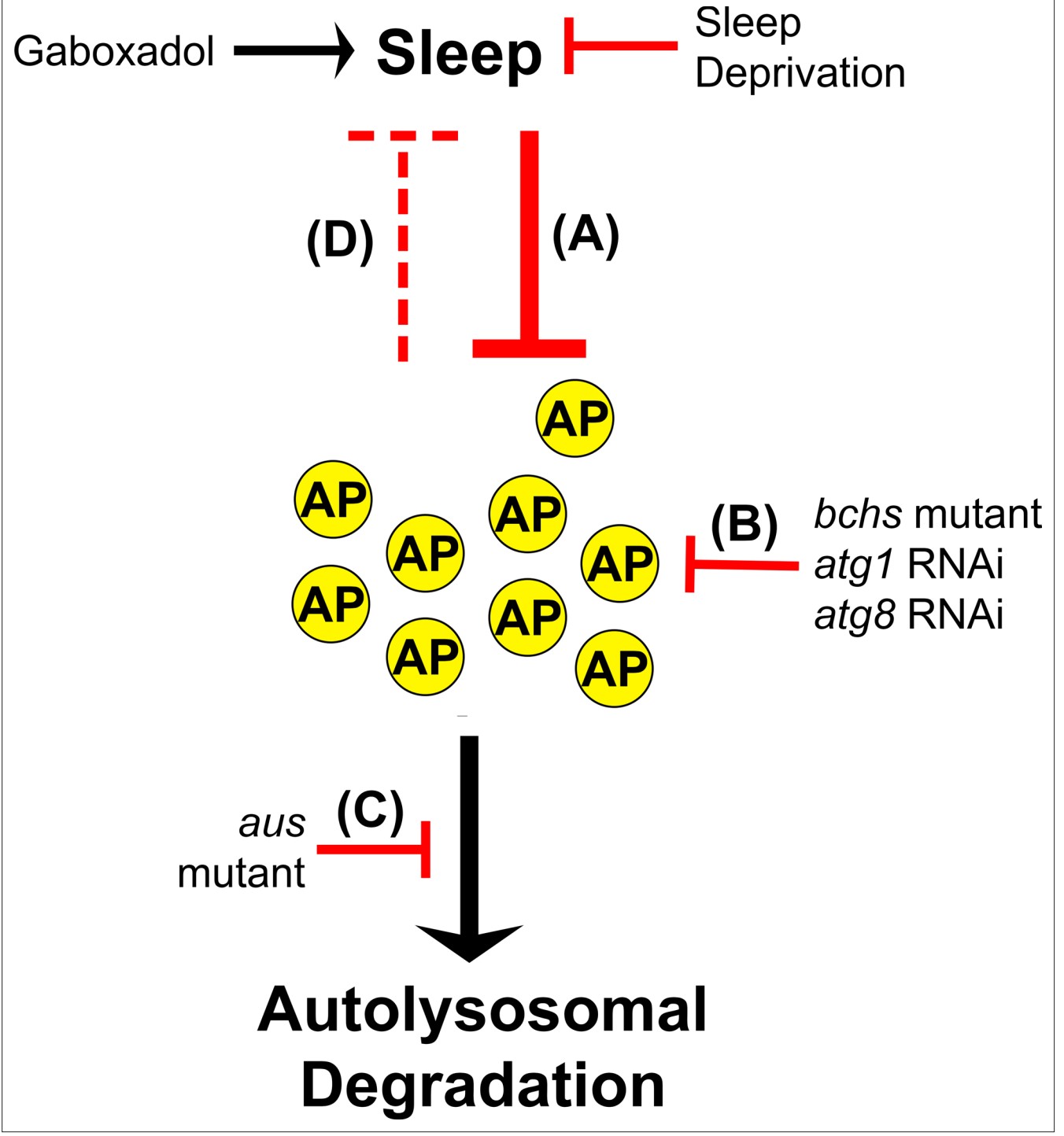

**Figure 9.** Model for sleep-autophagy interaction. This schematic details our model for how sleep and macroautophagy interact, based on our results. (**A**) Sleep decreases autophagosome number under normal conditions, in a manner that is sensitive to both gaboxadol gain or SD loss of sleep lasting between 11 and 14 hr. (**B**) The mutant *blue cheese*, pan-neuronal RNAi for atg1, and whole-fly RNAi for atg8b (suppressing both 8a and 8b homologs) are all known to inhibit autophagosome formation, and all increase sleep. (**C**) Neuronal loss-of-function in the aus mutant inhibits autophagosome degradation, and decreases sleep in a manner that is rescued by blocking autophagosome formation upstream. (**D**) The wake-promoting / sleep-inhibiting effects of autophagosome number are able to drive sleep behavior when strongly and sustainably adjusted by our genetic manipulations, but are unable to drive sustained waking after a single night of SD, as acute rebound sleep is well established to occur after sleep deprivation on this timescale. Together, this suggests that autophagosome inhibition of sleep is considerably weaker than sleep inhibition of autophagosome accumulation, with autophagosome number only becoming a strong enough signal to control sleep behavioral output with a very strong and/or sustained stimulus.

pathway in *Drosophila melanogaster* (*Kuhn et al., 2015*; *Thurmond et al., 2019*; *Gaudecker, 1963*). *aus* expression is also spatially enriched in tissues with high levels of developmental and adult autophagy, including brain, gut, and fat body (*Thurmond et al., 2019*). Finally, this hypothesis is consistent with our bioinformatic predictions, as the cell membrane, endoplasmic reticulum, and Golgi apparatus cellular compartments are all proposed donors of autophagosome membranes (*Supplementary file 2*; *Nishimura and Tooze, 2020*). We propose that Aus is required for transition of autophagosomes to autolysosomes, and so in its absence, autophagosomes accumulate.

To better understand the effect of autophagy disruptions on sleep, we exploited the vast existing mutant and RNAi resources available in *Drosophila* to conduct directed screening. This demonstrated sleep gain in multiple scenarios that impede the production of autophagosomes, including homozygotes for *bchs58* (*Figure 6A*), a loss-of-function mutant known to impair autophagosome maturation (*Sim et al., 2019*), and RNAi knockdown of a number of genes involved in autophagy initiation, autophagosome nucleation, and autophagosome maturation, in particular *atg1* and *atg8a/b* (*Figure 7* and *Figure 7—figure supplement 1*, *Figure 7—figure supplement 2*). Using a subset of these tools that were too weak to drive sleep gain in wild-type flies, we find complete suppression of *aus/+* sleep loss by *bchs/+*, and rescue of *aus/+* and *aus/aus* sleep loss by pan-neuronal RNAi for either *atg5* or *atg7* (*Figure 6B–D*). While the individual genes each have roles in additional pathways, the simplest explanation of consistent *aus* sleep rescue by three distinct autophagy gene loss-of-functions is that the observed autophagosome accumulation in *aus* mutants is a contributor to their short-sleeping phenotype (*Figures 5 and 6*). Together, our findings of abnormal sleep in both the *aus* and *bchs* autophagy mutants, as well as knockdown effects of functionally related clusters of canonical autophagy genes, demonstrate that strongly and sustainably disrupting autophagy in adulthood perturbs sleep, such that high autophagosome levels decrease sleep, while low autophagosome levels increase sleep (*Figure 9B–D*).

We then set out to determine whether this relationship reflects changes normally seen over the sleep-wake cycle. We found that Atg8a(+) autophagosomes accumulate during the waking hours and decrease during sleep, with autophagosome levels at daybreak increased by SD the preceding night, and autophagosome levels at nightfall decreased by gaboxadol-induced sleep the prior day (*Figure 8*). This demonstrated that at least one of two possibilities must be true in the absence of perturbations of autophagy: (i) sleep increases clearance of autophagosomes, and/or (ii) sleep decreases production of autophagosomes (*Figure 8*). Given the lack of effect on autophagosome/autolysosome ratio in our daybreak/nightfall, SD, and gaboxadol experiments, our data are most consistent with sleep reducing autophagosome production (*Figure 9A*). An attractive etiological explanation for this observation is elevated metabolic activity during wake generating waste that could enhance the production of autophagosomes, leading to autophagosome accumulation that is then run down over extended sleep. That said, we cannot fully rule out autophagosome clearance changes that occur as a gradual or late-onset feature of sleep. Sleep enhancement of the degradation of debris and damaged cells (*Singh and Donlea, 2020*) and the flushing of degraded wastes from the brain (*Artiushin et al., 2018*; *Xie et al., 2013*) seem to hint at a role for sleep in autophagosome clearance, and we believe that this possibility deserves further study. Finally, our findings in wild-type brains likely generalize to mammals, as various endosome-autophagosome-lysosome axis puncta accumulate in mice under chronic sleep fragmentation (*Xie et al., 2020*).

Regardless of whether autophagosome production, clearance, or both are affected during the normal sleep-wake cycle, our imaging data clearly show that waking increases and sleeping decreases autophagosome number (*Figure 8*). This is surprising given our mutant and RNAi data demonstrating that autophagosome level inhibits sleep (*Figures 5–7*). The most parsimonious synthesis of these results is a model in which sleep inhibits autophagosome formation on a 24 hr timescale (*Figure 9A*), while strong and/or sustained upregulation or downregulation of autophagosome levels is required to meaningfully modify sleep (*Figure 9B–D*). At present, it is unclear whether the small, transient fluctuations we observed in autophagosome number during a typical single day:night cycle are able to feed back on sleep regulation, or whether sleep unidirectionally regulates autophagy when their relationship is not perturbed by other factors.

This relationship between sleep and autophagy nonetheless has interesting implications for pathology. For instance, maladaptive autophagy flux provides a potential mechanism that could reinforce sleep loss in chronic sleep loss disorders. While a single night of SD causes only a modest

elevation of neuronal autophagosome number (*Figure 8E–H*), over many daily cycles chronic sleep loss could establish a deleterious positive feedback loop, with cumulative accumulation of autophagosomes becoming a strong enough wake-promoting cue to further suppress sleep.

This could also represent a mechanism coupling sleep loss disorders to increased incidence of neurodegeneration, and a driver of progressively worsening sleep disturbance noted over the course of many neurodegenerative disorders (*Winer and Mander, 2018*). Neurodegenerative disorders such as Alzheimer's disease are characterized by aggregating pathological proteins that strongly inhibit the autolysosomal clearance of autophagosomes (*Lee et al., 2010*; *Ling et al., 2009*; *Nixon et al., 2005*). Chronically elevated autophagosome levels in this context may disrupt sleep much like *aus*, and chronic sleep loss may in turn exacerbate autophagosome accumulation and further depress sleep, again forming a deleterious feedback loop that in this case begins with extended perturbation of autophagy rather than sleep.

This then begs the etiological question of why the sleep system would evolve such a potentially disastrous feedback loop with the autophagy pathway, one of its regulated outputs. The particularly strong sleep-promoting effects of neuronal *atg1* knockdown provide a potential clue (*Figure 7A, C and D*). Starvation is a well-known inducer of Atg1-dependent autophagy, and food scarcity is a common situation in nature that calls for both high levels of autophagy and suppression of sleep to allow for foraging (*Chang et al., 2009*; *Erion et al., 2012*; *Hale et al., 2013*). Thus, under starvation, the relationship we report for sleep and autophagosome levels would be adaptive. Given that the *aus* sleep-loss phenotype traces at least in part to neuropeptidergic populations, which are implicated in autophagy and also in behaviors such as feeding and sleep in *Drosophila* (*Bhukel et al., 2019*; *Dubowy and Sehgal, 2017*; *Dus et al., 2015*; *Lieberman et al., 2020*; *Melcher et al., 2007*), these populations may be particularly important for integrating homeostatic phenomena via changes in autophagosome levels. Examples of homeostatic integration are provided by findings that food-motivated learning in fruit flies is disrupted on a high-calorie diet, and that sleep is uncoupled from *Drosophila* memory consolidation by starvation (*Chouhan et al., 2021*; *Zhang et al., 2015*).

Alterations in autophagosome level, or perhaps contents, could integrate external and internal nutritive cues and differentially promote coupling of learning and memory to the food and/or sleep homeostats based on the fly's needs in a given situation. Indeed, it is tempting to speculate that under conditions of low sleep need and autophagosome level, autophagosomes may be important for clearing waste and maintaining overall cellular health, while very strong or prolonged disruptions to sleep or autophagy constitute a stress response able to modify behavior to adapt to environmental conditions. While we focus on the sleep homeostat in this manuscript, sleep's link to autophagy may be important for integrating sleep with not just the feeding homeostat, but also circadian rhythms and other biological drives more generally. Indeed, the well-documented involvement of autophagy in a range of nutritive, maintenance, stress-response, developmental, and other cellular functions could potentially position it as a cell-autonomous integrator of homeostatic needs writ large.

## Materials and methods
### Fly stocks
The *argus* mutant line was obtained in a chemical mutagenesis screen as described previously (*Shi et al., 2014*). Several *Drosophila* lines used to interrogate the *argus* allele, including aus2k-Gal4, both *cg16791* over-expression lines, and the *cg16791* Crispr mutant, were developed by our laboratory (see below). Mapping stocks, insertion mutants, some RNAi, and Gal4 lines were acquired from the Bloomington *Drosophila* Stock Center at Indiana. Other RNAi lines were acquired from the Vienna *Drosophila* Resource Center in Austria or the Kyoto Stock Center in Japan. See *Supplementary file 1*, Tab1 and *Supplementary file 3*, Tab 1 for details including stock center ID, genetic background, and figure-by-figure breakdown of all *Drosophila* lines used in this manuscript.

### Behavioral analysis
Flies were housed individually in glass tubes in Percival incubators. Beam-break activity was recorded with the Trikinetics DAM system (http://www.trikinetics.com/). Pysolo (http://www.pysolo.net) and custom Matlab software were used to analyze and plot sleep patterns (*Gilestro and Cirelli, 2009*; *Hsu et al., 2020*). All flies were entrained prior to and maintained on a 12 hr:12 hr light:dark cycle

for all behavior experiments, except where otherwise noted. Most behavior experiments examined behavior in flies that were ~3–5 days old at the start of recording for durations of up to 6 days, except where otherwise stated. Behavior experiments including homozygous *aus* groups examined sleep in flies of all groups that were ~3–7 days old at the start of recording for durations of up to 6 days. We expanded the acceptable age range for experiments including *aus* to allow us to maximize collections from a number of crosses with the *aus* allele that yielded few progeny.

## Mapping the *argus* locus

Classical genetic mapping with phenotypic markers was conducted for the *argus* allele very similarly to how we previously isolated *redeye* (*Shi et al., 2014*). The minimal overlap narrowed down the location of *argus* to the region distal of *ebony*.

SNP mapping: Genomic DNA of homozygous recombinants was subject to SNP analysis. SNP19M and SNP24M primers (*Supplementary file 1*, Tab2) were used for PCR amplification, and identified nucleotide polymorphism between wild type and the marker line. Scoring of recombinant progeny for *aus* further narrowed the locus to a ~ 5 M bases region between SNPs.

Deep Sequencing: Illumina paired-end DNA library kit was used to make genomic DNA libraries of *iso31* and *aus* homozygotes. The libraries were amplified ten times through PCR prior to Illumina Hi-Seq analyses. SNP calling algorithm identified polymorphisms.

## Molecular cloning

*Aus* promoter Gal4 constructs: Aus2kGal4 primers were used to amplify the *aus* 2 kb promoter region from genomic DNA derived from *iso31*, and cloned into pBPGw (Addgene #17574). *aus* cDNA clones: UAS-*aus* primers were used to amplify a truncated *aus* CDS from an *iso31* cDNA library and cloned into a pUAST-attB vector. UAS-*aus*FL primers were used to amplify the full-length *aus* cDNA from an *iso31* cDNA library and cloned into a pUAST-attB construct.

The PhiC31 integration system was adapted to target ausP-Gal4 constructs or UAS-*aus*(cDNA) constructs onto attP40 site on the 2nd chromosome or attP2 site on the 3rd chromosome.

Two gRNAs designed to generate a *CG16791/aus* knockout allele using the CRISPR/Cas9 system were cloned into pCFD4 (Addgene#49411) (*Port et al., 2014*). Separate primer sets were used to amplify and verify the target sequence. gRNA and primer sequences in *Supplementary file 1*, Tab 2. pHD-DsRed-attp-CG16791 vector: Approximately 1 kb upstream and downstream of the *argus* gene (CG16791) were PCR amplified using iso31 genomic DNA as a template. The 5' CG16791 arm was PCR amplified. The PAM sequence CCG inside the 5' arm was changed to <u>G</u>CG in the reverse primer (see underline) to prevent potential cutting by Cas9. The 3' CG16791 arm was amplified by cloning primers, and the PAM sequence inside 3' arm was changed from CCT to GCT using PAM elimination primers to prevent potential cutting by Cas9. PCR products of the 5' and 3' CG16791 arms were cloned into a SmaI site in the pBS-KS vector. After the construct was confirmed by sequencing with T7 and T3 primers, 5' and 3' arms were processed with AarI and SapI restriction enzymes, respectively, and inserted into AarI and SapI sites in pDsRed-attP (Addgene#51019). See *Supplementary file 1*, Tab2 for primer sequences.

The pCFD4-CG16791 gRNAx2 vector and pHD-DsRed-attp-CG16791 vector were mixed to final concentrations of 0.1 μg/μl and 0.5 μg/μl, respectively and injected into *vas*-Cas9 embryos by the Rainbow transgenic service. A single G0 male was crossed with Chr3 balancer virgin females to establish the line. Only G1 flies expressing DsRed in the eye were tested by extraction of gDNA followed by PCR. Further confirmation was done by southern blotting. The correct gene targeting lines were saved for testing in behavior assays. Knock-out flies (CG16791KO) were back-crossed with the *iso31* strain for several times and tested for behavior.

## Nucleic acid extraction and analysis

DNA Isolation: Flies (~15) were homogenized in DNA extraction buffer (100mMTris pH7.5; 100 mM EDTA; 100 mM NaCl; 0.5 % SDS). gDNAs were then isolated by sequential LiCl/KAc and isopropanol precipitations, and resuspended in TE for subsequent analysis.

Southern blot analysis: Roche Digoxin kit (Cat# 11093657910) was used to label DsRed DNA probes generated by PCR, using primers recorded in *Supplementary file 1*. Genomic DNA was digested with restriction enzymes and separated on 1% agarose gel before transfer to a nylon membrane. Digoxin

labeled probe was hybridized with the membrane at 42°C overnight. After washing, the membrane was exposed with a chemi-luminescence reaction through anti-Digoxin conjugated alkaline phosphatase (Cat# 11093274910).

RNA: Adult fly heads (~15) were subject to Trizol extraction (Ambion). High-capacity cDNA reverse transcription kits (Applied Biosystems) were used to make cDNA libraries.

Autophagy RNAi Screen for Sleep Behavior actinGS+ dicer and nsybGS+ dicer were separately crossed to RNAi's for genes with known roles in autophagy (*Supplementary file 3*). We initially measured total sleep in up to 16 female flies on 5% sucrose-agar food laced with 500 uM Sigma-Aldrich mifepristone / RU486 (Cat#: M8046) in ethanol vehicle (RU+ food), averaging sleep across days 4–5 of exposure to drug. Crosses with a median total sleep two hours or more higher or lower than both GS+ dicer and RNAi controls were considered possible hits (primary criterion; *Supplementary file 3*). In cases where different RNAi's for the same gene gave initial hits of opposite direction, we excluded both to avoid probable RNAi off-target effects (secondary criterion; *Supplementary file 3*). Finally, remaining possible hits were re-run a second time, measuring sleep under the same conditions as the initial screen. Crosses statistically different in the same direction from both controls in the combined runs (tertiary criterion) were considered RNAi hits (*Figure 7—figure supplement 1*).

For individual genes with multiple consistent RNAi hits (*atg1* and *atg8b*), we backcrossed five generations to *iso31*, then ran a more detailed analysis of sleep in females on both RU+ and ethanol vehicle laced (RU-) food, using the same crosses that gave hits in our screen. To confirm knockdown of target transcripts, we also crossed these alleles to actinGS+ dicer and harvested RNA from pools of 5 RU+ fed whole female flies with a Qiagen RNeasy Miniprep Plus kit (Cat# 74134). gDNA was removed by both included eliminator columns, and on-column Qiagen RNase-free DNAse treatment (Cat# 79254). RNA was reverse transcribed with Lifetech Superscript II Reverse Transcriptase (Cat# 18064071). cDNAs for putative RNAi target genes and *alpha-tubulin* were amplified using Lifetech SYBR Green PCR mix (Cat# 4364346) and primers in *Supplementary file 1* on an Applied Biosystems ViiA7 qPCR machine. We calculated relative transcript levels by ddCT.

## Live imaging experiments

Brains from approximately 1 week old adult female flies singly housed on our lab's standard yeast-molasses food were dissected and mounted in chilled artificial hemolymph (108 mM NaCl; 5 mM KCl; 2 mM $CaCl_2$; 8.2 mM $MgCl_2$-$6H_2O$; 4 mM $NaHCO_3$; 1 mM $NaH_2PO4$-$H_2O$; 5 mM trehalose; 10 mM sucrose; 5 mM HEPES; 265mOsm and pH7.5) (*Cohn et al., 2015*). They were live imaged embedded in vacuum grease with a 40 X water immersion objective at 1.3 X digital zoom under a Leica confocal microscope at Alexa488 (green) and Alexa594 (red) wavelengths. Z-stacks containing ~60 μm of the central brain starting from the tips of the antennal lobes were captured.

Ilastik machine learning software was trained to isolate all mCherry(+) puncta from our Z-stacks (*Berg et al., 2019*). Briefly, for each experiment an equal number of representative brains from each group were marked for signal and noise in the red channel by a human scorer to train the Ilastik algorithm. Slices from the front, middle, and back of each stack were used, taking care to mark a range of diverse examples of signal and background. A similar number of markings were made between groups, to avoid biasing the algorithm. Ilastik's prediction of signal and background was then reversibly overlaid on unmarked sample sections and visually inspected for accuracy by the human scorer. Once the algorithm passed inspection, simple segmentations of all brains were generated by Ilastik to define puncta and background for input into ImageJ. ImageJ was then used to measure mCherry(+) puncta count and size, and each mCherrry puncta's green channel intensity from GFP. We then thresholded to background GFP intensity within each brain, and counted mCherry(+) puncta with green fluorescence intensity exceeding background to determine autophagosome and autolysosome percentages. The Ilastik algorithm was validated by quantifying autophagy following treatment with the autophagy-inducer rapamycin (see below).

Brains for Ilastik validation were incubated in artificial hemolymph supplemented with either 2 μM LC Laboratories rapamycin (Cat#: R5000) in ethanol vehicle, or ethanol vehicle alone, for ~2 hours prior to imaging. The drug condition was maintained for each group throughout imaging.

For sleep deprivation, flies were placed in DAM monitors in locomotor tubes filled with fresh yeast-molasses food on top of a mechanical deprivator. During the night preceding imaging, flies were shaken for a period of 2 s every 20 s to disrupt sleep, as previously described (*Toda et al., 2019*).

For sleep induction, flies were flipped from regular yeast-molasses food onto yeast-molasses food supplemented with either Sigma-Aldrich gaboxadol hydrochloride (Cat#: T101) in water vehicle diluted to 0.1 mg/mL final concentration, or water vehicle alone, during ZT0-1. Flies were maintained on the supplemented food for ~12 hr before imaging from ZT12-14. Sleep was recorded for at ~11 hr after flip onto drugged food, to verify that we observed gaboxadol-induced sleep gain as previously described in the same flies whose brains were imaged (*Berry et al., 2015*; *Dissel et al., 2015*).

## Statistics

Statistics were run in GraphPad Prism or JMP software. Shapiro-Wilkes tests were used to assess normality of each group for each individual experiment. Multiple-comparison correction was appropriately applied where multiple comparisons tested multiple hypotheses, but not where multiple comparisons were made to test a single hypothesis, as in non-geneswitch Gal4 driven RNAi and rescue experiments conducted in the manuscript (*Shaffer, 1995*).

## Acknowledgements

We thank Han Wang and Zhifeng Yue for assistance with fly work. We also thank Ana Maria Cuervo for helpful suggestions regarding autophagy measurements and Taichi Hara, Toshifumi Tomoda, and Nobuo Noda for critical comments on this manuscript.

## Additional information

### Competing interests

Amita Sehgal: Reviewing editor, eLife. The other authors declare that no competing interests exist.

### Funding

| Funder | Grant reference number | Author |
|---|---|---|
| Howard Hughes Medical Institute | | Amita Sehgal |
| National Institutes of Health | F32AG056081-03 | Joseph L Bedont |
| National Institutes of Health | K99NS118561-01 | Joseph L Bedont |

The funders had no role in study design, data collection and interpretation, or the decision to submit the work for publication.

### Author contributions

Joseph L Bedont, Conceptualization, Data curation, Formal analysis, Funding acquisition, Investigation, jlb was the first author primarily responsible for identifying bchs long sleep and its suppression of aus; all live imaging experiments in the manuscript; the autophagy rnai sleep screen; and the aus rnai sleep experiments with nsybgal4 and actings drivers, Methodology, Visualization, Writing – original draft, Writing – review and editing; Hirofumi Toda, Conceptualization, Data curation, Formal analysis, ht and ms were the first authors jointly primarily responsible for isolating the aus short sleep mutant, Methodology, Visualization, Writing – original draft, Writing – review and editing, cloning the aus gene and mapping aus sleep to peptidergic neurons, cloning the aus gene and mapping aus sleep to peptidergic neurons. ht was also the first author primarily responsible for demonstrating atg5 and atg7 rescue of aus short sleep; Mi Shi, Conceptualization, Data curation, Formal analysis, ms and ht were the first authors jointly primarily responsible for isolating the aus short sleep mutant, Methodology, Visualization, Writing – original draft, Writing – review and editing, cloning the aus gene and mapping aus sleep to peptidergic neurons; Christine H Park, Data curation, Formal analysis, Supervision, under ht's supervision; Christine Quake, cq contributed to isolating the aus short sleep mutant and cloning the aus gene, Data curation; Carly Stein, cs contributed to portions of the autophagy rnai screen, Data curation, Investigation, and some live imaging experiments; Anna Kolesnik, ak contributed to portions

of the autophagy rnai screen and some live imaging experiments, Investigation, Methodology, and some live imaging experiments; Amita Sehgal, Conceptualization, Funding acquisition, Project administration, Supervision, Writing – original draft, Writing – review and editing

### Author ORCIDs
Joseph L Bedont  http://orcid.org/0000-0002-1614-4805
Hirofumi Toda  http://orcid.org/0000-0002-6247-2826
Mi Shi  http://orcid.org/0000-0002-3044-912X
Amita Sehgal  https://orcid.org/0000-0001-7354-9641

### Decision letter and Author response
Decision letter https://doi.org/10.7554/eLife.64140.sa1
Author response https://doi.org/10.7554/eLife.64140.sa2

---

## Additional files

### Supplementary files
• Supplementary file 1. Lines and Primers. Tab 1: A figure-by-figure breakdown of alleles, sources, and backgrounds for each fly line used in most figures of the manuscript. Tab2: A list of all primer sequences used in producing and validating the novel fly lines described in the manuscript.

• Supplementary file 2. Bioinformatic analysis of the CG16791/ Aus protein product. An unbiased ProDom analysis of the full-length CG16791, Isoform A protein sequence identified a number of candidate transmembrane domains. Validation with TMPred produced a similar 5-transmembrane best-fit topological prediction for all naturally occurring isoforms of CG16791, as well as our UAS-aus construct protein product. Deep-Loc-1.0 predicted the cell membrane as the most likely initial insertion site for all of these same CG16791 sequences, with the endoplasmic reticulum and Golgi apparatus as possible alternative insertion sites.

• Supplementary file 3. Autophagy RNAi Screen, First-Pass Sleep for All Crosses. Tab 1: A list of all RNAi's used in the screens, including unambiguous stock center IDs. Tab 2: First-pass medians, interquartiles, and n's for total sleep in females on RU+ food for each nsybGS> dcr,RNAi cross with appropriate controls. Crosses that passed primary criterion are indicated, and annotated with whether they passed subsequent criteria or not and, if not, why. Tab 3: First-pass medians, interquartiles, and n's for total sleep in females on RU+ food for each actinGS> dcr,RNAi cross with appropriate controls. Crosses that passed primary criterion are indicated, and annotated with whether they passed subsequent criteria or not and, if not, why.

• Transparent reporting form

### Data availability
All data generated or analysed during this study are included in the manuscript and supporting files.

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
