## [Decision Letter]

**Acceptance summary:**

Using mutagenesis screen, classical mapping, and genomic sequencing, Toda et al. identified a novel short-sleeping *Drosophila* mutant, argus, which also exhibits increased accumulation of autophagosomes. This finding was examined in contrast to a long-sleeping mutant, blue cheese, which exhibits impaired autophagosome maturation. The authors also showed that autophagosomes accumulate due to sleep deprivation, providing evidence for a bidirectional relationship between sleep and autophagy. These exciting results identify autophagy as a potential function and regulator of sleep.

**Decision letter after peer review:**

Thank you for submitting your article "Short and long sleeping mutants reveal links between sleep and macroautophagy" for consideration by *eLife*. Your article has been reviewed by 2 peer reviewers, and the evaluation has been overseen by K VijayRaghavan as the Senior and Reviewing Editor. The reviewers have opted to remain anonymous.

The reviewers have discussed the reviews with one another and the Reviewing Editor has drafted this decision to help you prepare a revised submission.

We would like to draw your attention to changes in our policy on revisions we have made in response to COVID-19 (https://elifesciences.org/articles/57162). Specifically, when editors judge that a submitted work as a whole belongs in *eLife* but that some conclusions require a modest amount of additional new data, as they do with your paper, we are asking that the manuscript be revised to either limit claims to those supported by data in hand or to explicitly state that the relevant conclusions require additional supporting data.

Summary:

Using forward genetic mutagenesis (EMS) screen, classic mapping, and genomic sequencing, Toda et al. identified a novel short-sleeping *Drosophila* mutant, argus, which also exhibits increased accumulation of autophagosomes. This finding was examined in contrast to a long-sleeping mutant, blue cheese, which exhibits impaired autophagosome maturation. The authors also showed that autophagosomes accumulate due to sleep deprivation, providing evidence for a bidirectional relationship between sleep and autophagy. These results identify autophagy as a potential function and regulator of sleep. The work is very exciting and experiments are well done, but important concerns remain and have been outlined below.

Essential revisions:

1. The authors link regulation of autophagy directly to regulation of sleep using two non-standard regulators of autophagy (aus and bch), for which autophagic phenotypes may be an indirect consequence of defects in other functions. While it is true that knockdown of known autophagic regulators can compensate for their dysfunction and the sleep phenotype, knockdown of these autophagic regulators does not itself affect sleep in ways that fit their hypothesis. The authors show the correlation of autophagosome accumulation with sleep and sleep deprivation; however, it is unclear if sleep-related autophagy is distinct from circadian-regulated autophagy (ie, Ma et al., 2011) and if sleep deprivation-induced autophagy is distinct from starvation-induced autophagy. Moreover, if the authors' hypothesis is correct, then RNAi of autophagy components (Atg5, 7) should alter the autophagosome phenotype as well as the sleep phenotype. That said, these data are extremely exciting and provide a clear correlation between autophagy and sleep regulation.

However, the major concern is that there are mainly sleep phenotypes from manipulating autophagy in a disease context so it's not clear how relevant autophagy is for normal sleep. Metaphorically, If someone robs a bank and drives away, the fact that messing up all the stoplights in the city may prevent their escape doesn't necessarily tell us how law enforcement is supposed to work. That said, a mechanism connecting autophagy and sleep is highly appealing and would make sense in the context of the field. Therefore at least two more experiments could really tell us if autophagosome accumulation itself is important for normal sleep regulation. The authors should see if they can be done and if not feasible in a reasonable time, discuss the above concerns substantively and tone down the conclusions accordingly.

2. The first experiment is to look at autophagosome number over the missing parts of the circadian cycle in wild-type flies and, ideally, in the aus and/or bch mutants. In Ma et al., 2011, six time points from the mouse liver clearly show that liver autophagosomes accumulate during the day and decrease at night. Looking at other parts of the circadian cycle is critical to testing their model. If autophagosomes also accumulate during the day in fly brains, this would suggest that the act of decreasing (not increasing) autophagosomes causes sleep. This would make much more sense in the context of their findings. (Though would have to be explained regarding mouse sleep…) If, on the other hand, autophagosomes accumulate during the night in fly brains, this would argue for their model (as we understand it) in which autophagosome accumulation promotes sleep and excess accumulation "breaks" the homeostatic. The model itself is somewhat counter-intuitive and is not easy to reconcile with the Atg RNAi phenotype but we can accept that this may be happening in this case.

3. The second experiment really needed here is something that pins down the role of autophagy in normal sleep. There are many acceptable options for this point. If experiment 1 above shows that autophagosomes continue to accumulate and peak during the waking state and the decrease in autophagosomes (inhibition of autophagy?) triggers sleep, then the Atg7 RNAi (alone, not in disease context) makes a little more sense and these should have low autophagosome formation and therefore sleep more. If, on the other hand, the authors see the opposite result in experiment 1, it would be good to either test other classic autophagy mutants/RNAi for sleep phenotype or overexpress Atg1 and induce autophagy (ideally in a timed manner using an inducible driver such as elav geneswitch or heat-induced promoter) and examine the effects on sleep and autophagosome number. As a note, starvation induces autophagy (presumably increasing autophagosomes) and causes foraging behavior and sleep loss. This is consistent with their current model and, if true, then the prediction falls out that starvation of Atg5 and Atg7 RNAi mutants will not induce sleep loss.

The key for this model may be autophagy flux… that is, the flux through the autophagy pathway. There are GFP reporters for different types of autophagy targets and one can perform pulse-chase assays, monitoring GFP levels by microscopy or western blot. These experiments might be much more informative than static pictures at only two time points.

---

## [Author Response]

Essential revisions:1. The authors link regulation of autophagy directly to regulation of sleep using two non-standard regulators of autophagy (aus and bch), for which autophagic phenotypes may be an indirect consequence of defects in other functions. While it is true that knockdown of known autophagic regulators can compensate for their dysfunction and the sleep phenotype, knockdown of these autophagic regulators does not itself affect sleep in ways that fit their hypothesis.

We believe that rescue of *aus* short-sleep by *bchs*, and by knockdown of canonical autophagy genes, *atg5* and *atg7*, together demonstrates that blockade of autophagosome degradation drives *aus* short-sleep. That said, we acknowledge the reviewer’s point and have addressed it systematically by including a sleep screen of 135 RNAis covering 44 genes involved in autophagy. By expressing these under the control of pan-neuronal *nsyb*-gene switch (pan-neuronal, medium strength) or actin-gene switch (whole-fly, extremely strong) drivers, we were able to acutely induce knockdown in adulthood. We find that a number of RNAis, functionally clustered around initiation of autophagy and maturation of autophagosomes up to the point of Atg8 incorporation, increase sleep. This includes sleep increases with multiple distinct RNAis for *Atg1* and *Atg8b* (targets both *8a* and *8b* homologs).

The same screen showed that expression with the temporally drug-gated actin-gene switch driver increases baseline sleep with the same *atg7* RNAi allele we used to rescue *aus* sleep loss with the elav-Gal4 driver. The reduced sleep phenotype of elavGal4>*atg7* knockdown alone was most likely due to developmental effects, which nevertheless was helpful for our purposes because rescue of *aus* by a manipulation that increased sleep by itself might have been interpreted as an additive effect.

In sum, the addition of this behavioral screen considerably strengthens the evidence supporting our hypothesis that perturbing autophagosome number can affect sleep. Courtesy of the many gain-of-sleep phenotypes we identify in our RNAi screen, we can now be reasonably confident that the previous lack of baseline sleep gain with canonical autophagy RNAis is due to the vagaries of RNAi knockdown efficiency and/or timing, rather than a refutation of our model.

The authors show the correlation of autophagosome accumulation with sleep and sleep deprivation; however, it is unclear if sleep-related autophagy is distinct from circadian-regulated autophagy (ie, Ma et al., 2011) and if sleep deprivation-induced autophagy is distinct from starvation-induced autophagy. Moreover, if the authors' hypothesis is correct, then RNAi of autophagy components (Atg5, 7) should alter the autophagosome phenotype as well as the sleep phenotype. That said, these data are extremely exciting and provide a clear correlation between autophagy and sleep regulation.However, the major concern is that there are mainly sleep phenotypes from manipulating autophagy in a disease context so it's not clear how relevant autophagy is for normal sleep. Metaphorically, If someone robs a bank and drives away, the fact that messing up all the stoplights in the city may prevent their escape doesn't necessarily tell us how law enforcement is supposed to work. That said, a mechanism connecting autophagy and sleep is highly appealing and would make sense in the context of the field. Therefore at least two more experiments could really tell us if autophagosome accumulation itself is important for normal sleep regulation. The authors should see if they can be done and if not feasible in a reasonable time, discuss the above concerns substantively and tone down the conclusions accordingly.2. The first experiment is to look at autophagosome number over the missing parts of the circadian cycle in wild-type flies and, ideally, in the aus and/or bch mutants. In Ma et al., 2011, six time points from the mouse liver clearly show that liver autophagosomes accumulate during the day and decrease at night. Looking at other parts of the circadian cycle is critical to testing their model. If autophagosomes also accumulate during the day in fly brains, this would suggest that the act of decreasing (not increasing) autophagosomes causes sleep. This would make much more sense in the context of their findings. (Though would have to be explained regarding mouse sleep…) If, on the other hand, autophagosomes accumulate during the night in fly brains, this would argue for their model (as we understand it) in which autophagosome accumulation promotes sleep and excess accumulation "breaks" the homeostatic. The model itself is somewhat counter-intuitive and is not easy to reconcile with the Atg RNAi phenotype but we can accept that this may be happening in this case.

We agree that ruling out circadian confounds for our manuscript is important. However, we respectfully disagree that simply measuring autophagy flux at more circadian timepoints is the best way to address this. In the absence of a homeostatic perturbation, the 24-hour pattern of autophagy posited by the reviewer as supportive of sleep control could still simply be a pattern driven by the circadian clock. We believe that examining the response of the autophagy system to homeostatic perturbation of sleep is the appropriate way to assess whether the sleep homeostat has a role distinct from the circadian clock. To this end, we now include both the previous-version SD experiment showing that loss of sleep at night increases daybreak Atg8a(+) puncta number, as well as a new gaboxadol experiment showing that pharmacologically enforced sleep during the day decreases nightfall Atg8a(+) puncta number (with no significant effects on AP/AL ratio or puncta size in either case). These results demonstrate a distinct role for the sleep homeostat.

Importantly, the results of the experiments above are consistent with our model. To be clear, we believe that the higher autophagosome number in the early night (ZT12-14, or 0-2 hours from lights off) reflects accumulation during the day. However, accumulation during the day does not argue for a decrease in autophagosomes causing sleep, as inferred by the reviewer, because: (1) the number is high in the early night when flies fall asleep, and (2) Prolonged wake (sleep deprivation) increases accumulation of autophagosomes.

Our mutant and RNAi data consistently suggest that strong and/or long-term elevation of autophagosome number inhibits sleep, while strong and/or long-term depression of autophagosome number promotes sleep. We thank the reviewers for their feedback on our model, as it led us to re-evaluate and arrive at what we believe is a more parsimonious and easier to understand synthesis of our data. As shown in our updated Figure 9, we believe that sleep regulates autophagy in a normal day/night cycle, and even under acute single-cycle manipulations of sleep such as in our mechanical SD and gaboxadol experiments. But effects of autophagy on sleep are only evident with a sufficiently strong and/or sustained change to autophagosome number, such as in the mutants we report here. Notably, all the mutant data support the hypothesis that a high/sustained decrease in autophagosomes promotes sleep and vice versa.

Finally, the reviewer’s comment suggests that we inadvertently gave the impression that we were proposing that sleep, NOT circadian rhythms, regulates autophagy in *Drosophila* neurons. This is not the case. Rather, we are asserting that the sleep homeostat plays a role (undoubtedly among other behavioral factors, including feeding and, likely, circadian rhythms) in regulating autophagy. To clarify this, we now explicitly state at the end of our discussion that sleep control of autophagy is not mutually exclusive with circadian and feeding control. Indeed, starvation and long-term circadian disruptions are two stimuli that may well be capable of driving autophagosome numbers low or high enough for a long period of time to exert an influence on sleep. The former possibility, in particular, we go into at some length elsewhere in the Discussion.

3. The second experiment really needed here is something that pins down the role of autophagy in normal sleep. There are many acceptable options for this point. If experiment 1 above shows that autophagosomes continue to accumulate and peak during the waking state and the decrease in autophagosomes (inhibition of autophagy?) triggers sleep, then the Atg7 RNAi (alone, not in disease context) makes a little more sense and these should have low autophagosome formation and therefore sleep more. If, on the other hand, the authors see the opposite result in experiment 1, it would be good to either test other classic autophagy mutants/RNAi for sleep phenotype or overexpress Atg1 and induce autophagy (ideally in a timed manner using an inducible driver such as elav geneswitch or heat-induced promoter) and examine the effects on sleep and autophagosome number. As a note, starvation induces autophagy (presumably increasing autophagosomes) and causes foraging behavior and sleep loss. This is consistent with their current model and, if true, then the prediction falls out that starvation of Atg5 and Atg7 RNAi mutants will not induce sleep loss.

We tested sleep behavior following RNAi knockdown of a large number of autophagy-relevant genes with inducible drivers and found results largely consistent with our hypothesis, as discussed at length above. There appears to be some confusion about our hypothesis, so to clarify, we find that autophagosomes increase with wakefulness and decline with sleep, potentially from accumulation of cellular waste during wake, which is then reduced during sleep. The new gaboxadol experiment, which shows a reduction in autophagosomes with induced sleep during the day (fruit flies’ normally active phase) supports this hypothesis. While this might suggest that an accumulation of autophagosomes drives sleep, we have no reason to believe that autophagy drives sleep in a daily cycle. On the other hand, a sustained change in autophagosome number affects sleep, but in the opposite direction from predictions based on the day:night profile, as demonstrated by *bchs* and *aus* mutant phenotypes and now supplemented by results of our RNAi-based sleep screen of autophagy regulators. All the mutant and RNAi phenotypes are consistent with a model in which a sustained increase in autophagosomes reduces sleep while a decrease increases sleep (Figure 9).

Such a persistent change in autophagosomes may occur in certain pathological conditions; for instance, the *aus* short-sleep with autophagosome accumulation situation is reminiscent of what is seen in neurodegenerative disorders including Alzheimer’s; interestingly, sleep is affected in these disorders. One notable ethological scenario where an effect of autophagosomes on sleep could be evolutionarily valuable is under conditions of food scarcity, where elevated autophagosome levels, required to release nutritional stores, need to be coordinated with extended waking to facilitate foraging.

We now more clearly define our model and explain its implications, in our model figure and Discussion sections.

The key for this model may be autophagy flux… that is, the flux through the autophagy pathway. There are GFP reporters for different types of autophagy targets and one can perform pulse-chase assays, monitoring GFP levels by microscopy or western blot. These experiments might be much more informative than static pictures at only two time points.

We agree that experiments like the ones proposed would be interesting; however, addressing the contents of the sleep-regulated autophagosomes with reporters for different targets is a distinct question from whether the sleep homeostat influences autophagy to begin with. It would also be a very laborious and time-consuming effort that could take years. We are unable to conduct such studies in a reasonable amount of time for resubmission of the present manuscript.